# Towards Representation Learning for Weighting Problems in Design-Based Causal Inference

**Oscar Clivio**[1]      **Avi Feller**[2]      **Chris Holmes**[1]

[1]Department of Statistics, University of Oxford
[2]Goldman School of Public Policy and Department of Statistics, University of California, Berkeley

## Abstract

Reweighting a distribution to minimize a distance to a target distribution is a powerful and flexible strategy for estimating a wide range of causal effects, but can be challenging in practice because optimal weights typically depend on knowledge of the underlying data generating process. In this paper, we focus on design-based weights, which do not incorporate outcome information; prominent examples include prospective cohort studies, survey weighting, and the weighting portion of augmented weighting estimators. In such applications, we explore the central role of representation learning in finding desirable weights in practice. Unlike the common approach of assuming a well-specified representation, we highlight the error due to the choice of a representation and outline a general framework for finding suitable representations that minimize this error. Building on recent work that combines balancing weights and neural networks, we propose an end-to-end estimation procedure that learns a flexible representation, while retaining promising theoretical properties. We show that this approach is competitive in a range of common causal inference tasks.

## 1 INTRODUCTION

Estimating causal effects is a fundamental task in multiple fields such as epidemiology [Westreich et al., 2017], medicine [Rosenbaum, 2012], public policy [Eli Ben-Michael and Jiang, 2024] or economics [Sekhon and Grieve, 2012]. Some challenges include removing the influence of confounders [Pearl et al., 2016] or generalizing a treatment effect estimated on a randomized control trial (RCT) to a target observational population [Degtiar and Rose, 2023, Colnet et al., 2024]. Weighting approaches, which target a causal effect as an expectation under a reweighting of the original distribution, can address many of these problems [Ben-Michael et al., 2021, Colnet et al., 2024, Johansson et al., 2022].

In this paper, we focus on finding so-called *design-based weights*, which do not incorporate any outcome information, either out of principle or out of necessity; as such, we cannot apply existing approaches involving outcomes off the shelf. Most prominently, design-based weights arise in the classical literature on the design of observational studies, which stresses the importance of separating the "design" and "analysis" phases of a non-randomized study [Rubin, 2008], and therefore stresses the importance of estimating weights without using the outcome. Such weights also arise in *prospective cohort studies* [Song and Chung, 2010] and in *survey design* [Lohr, 2021], in which researchers have not yet collected outcomes, as well as in applications in which it is useful to develop a single set of outcome-agnostic weights, such as in analyses with multiple outcomes of interest [Ben-Michael et al., 2024]. Finally, in doubly robust methods that combine outcome and weighting models, such as in Automatic Debiased Machine Learning (AutoDML) [Chernozhukov et al., 2022b] or augmented balancing weights [Ben-Michael et al., 2021], the weights are typically estimated without using outcomes.

Such methods for finding design-based weights generally rely on minimizing a probability distance between the weighted distribution and a reference distribution. The optimal distance, however, typically depends on the unknown data generating process (DGP). This has led to a large literature on learning an adequate *representation*, a mapping of the covariate space to another manifold, that retains important properties of the DGP. Standard representations include balancing scores [Rosenbaum and Rubin, 1983b], sufficient dimension reduction [Luo and Zhu, 2020], and variable selection [Brookhart et al., 2006]. The correctness of these representations typically relies on unverifiable assumptions and the analyst is left without guarantees on the bias of the weighting estimator if they are not met, leading to poor per-

formance in practice [Kang and Schafer, 2007]. More recent approaches learn a representation implicitly by, for example, modelling weights directly as neural networks, however they only provide guarantees on the bias for specific DGPs, e.g. when the outcome model is piece-wise constant [Ozery-Flato et al., 2018] or follows a neural network architecture [Kallus, 2020a]. Despite these advances, there are not currently principled procedures to directly assess and control the quality of a representation and its impact of the bias on the weighted estimator for any possible data generating process.

This constitutes our two main contributions. (1) We quantify the information lost by using a weighted estimator based on a representation, rather on the original covariates, through a "confounding bias" and a "balancing score error", and give guarantees on the resulting bias of the estimator for any (posited class of the) outcome model. (2) We develop a method inspired by DeepMatch [Kallus, 2020a] and RieszNet [Chernozhukov et al., 2022a] that learns such representations from data. Unlike the original RieszNet application, however, we do not incorporate outcome information. We show promising performance of this approach on benchmark datasets in treatment effect estimation. This learnt representation can serve as the input of any weighting method, making it a generic pre-processing method.

## 2 BACKGROUND

### 2.1 SETUP AND NOTATION

Let $P(X, \tilde{Y})$ be a **source** distribution on *covariates* $X$ and some *pseudo-outcomes* $\tilde{Y}$, and $Q$ be a **target** distribution on covariates. For any distribution $R$ and random variable $Z$, denote $R_Z$ the law $R(Z)$. We assume that we have access to (not necessarily disjoint) i.i.d. samples $\mathcal{P}$ from $P$ and $\mathcal{Q}$ from $Q$. Let $\mathbb{E}_R[Z]$ be the expectation of a random variable $Z$ under the distribution $R$. We call a **weight function wrt** $P$ or **weights wrt** $P$ any measurable $P_X$-a.s. non-negative function $w(x)$ of covariates such that $\mathbb{E}_P[w(X)] = 1$. Any weight function $w$ wrt $P$ induces a distribution $P^w$ such that $\frac{\mathrm{d}P_X^w}{\mathrm{d}P_X}(x) = w(x)$ and $P^w(\tilde{Y}|X) = P(\tilde{Y}|X)$, where we say that $P$ is **reweighted** by $w(X)$, with $\mathbb{E}_{P^w}[f(X)] = \mathbb{E}_P[w(X)f(X)]$ for any function $f$. Let $\mathbb{E}_P[\tilde{Y}|X = x]$ be a function of interest, which we call **the outcome model**. We are interested in the **target estimand** $\mathbb{E}_Q[\mathbb{E}_P[\tilde{Y}|X]]$. In general, we do not have access to either the outcome model or the target estimand. That said, for any weight function $w(x)$ wrt $P$, $\hat{\tau}_w := \frac{1}{|\mathcal{P}|} \sum_{i \in \mathcal{P}} w(X_i)\tilde{Y}_i$ is an unbiased estimator of $\mathbb{E}_{P^w}[\mathbb{E}_P[\tilde{Y}|X]]$ as soon as $\mathbb{E}_P[w(X)\tilde{Y}]$ is well-defined. All of this motivates our problem statement.

**Problem 2.1.** Find a weight function $w(X)$ wrt $P$ such that

$$\mathbb{E}_{P^w}\left[\mathbb{E}_P[\tilde{Y}|X = x]\right] = \mathbb{E}_Q\left[\mathbb{E}_P[\tilde{Y}|X = x]\right]$$

This generalizes many weighting problems in causal inference. Generally, let $A$ denote the treatment variable, and $Y$ denote the outcome. We assume that the values of $A$ belong to a finite space $\mathcal{A}$. For $a \in \mathcal{A}$, we denote $Y(a)$ the potential outcome wrt $a$, which is the realized outcome if the subject were to receive treatment $a$. In the context of transportability, we also introduce a binary indicator $S$ for membership in a RCT population, thus $A \perp\!\!\!\perp X|S = 1$ and $(Y(1), Y(0)) \perp\!\!\!\perp A|S = 1$. Let $P^{\text{data}}(X, Y, S, A, (Y(a))_{a \in \mathcal{A}})$ be the true data distribution. In the absence of subscript, we assume that the expectation operator is that wrt $P^{\text{data}}$, that is $\mathbb{E} := \mathbb{E}_{P^{\text{data}}}$. Then, Problem 2.1 can be applied to the following weighting problems (details in Appendix A):

- *Average Treatment Effect on the Treated (ATT).* The pseudo-outcome is $Y$; the source and target distributions are $P^{\text{data}}(X, Y|A = 0)$ and $P^{\text{data}}(X|A = 1)$, respectively; the outcome model is $\mathbb{E}[Y(0)|X = x]$; the estimand is $\mathbb{E}[Y(0)|A = 1]$.

- *Average Treatment Effect (ATE).* Let $a \in \mathcal{A}$ be fixed. The pseudo-outcome is $Y$; the source and target distributions are $P^{\text{data}}(X, Y|A = a)$ and $P^{\text{data}}(X)$, respectively; the outcome model is $\mathbb{E}[Y|A = a, X = x]$; the estimand is $\mathbb{E}[Y(a)]$.

- *Transportability.* The pseudo-outcome is

$$\tilde{Y} := \frac{AY}{P^{\text{data}}(A = 1|S = 1)} - \frac{(1 - A)Y}{P^{\text{data}}(A = 0|S = 1)};$$

the source distribution is the joint covariate and pseudo-outcome distribution in the RCT $P^{\text{data}}(X, \tilde{Y}|S = 1)$; the target distribution is the covariate distribution in the target population $P^{\text{data}}(X|S = 0)$; the outcome model is the conditional average treatment effect (CATE) $\mathbb{E}[Y(1) - Y(0)|X = x]$; the estimand is the ATE on the target population, $\mathbb{E}[Y(1) - Y(0)|S = 0]$.

One solution to these problems has the following form:

**Definition 2.2.** We call **true weights between** $P$ **and** $Q$ the Radon-Nikodym derivative $\frac{\mathrm{d}Q_X}{\mathrm{d}P_X}$, which is a weight function wrt $P$.

These weights are also known as *inverse probability weights* or the *Riesz representer* [Hirshberg and Wager, 2021, Chernozhukov et al., 2022b]. They are uniquely defined [Ben-Michael et al., 2021] by, for any measurable function $f$,

$$\mathbb{E}_P\left[\frac{\mathrm{d}Q_X}{\mathrm{d}P_X}(X)f(X)\right] = \mathbb{E}_Q[f(X)].$$

In particular, this holds for $f(x) = \mathbb{E}_P[\tilde{Y}|x]$, which solves Problem 2.1. In practice, the true weights $\frac{dQ_X}{dP_X}$ are unknown; we turn to estimating them and more generally obtaining solution weights in the next section.

Finally, to ensure that true weights are well-defined, we make the following assumption, which is equivalent to *overlap* in ATE estimation [Bruns-Smith et al., 2023] and *support inclusion* [Colnet et al., 2024] in transportability.

**Assumption 2.3.** $Q_X$ is absolutely continuous wrt $P_X$.

As we discuss in the introduction, we are in the setting where outcomes $Y_i$ and pseudo-outcomes $\tilde{Y}_i$ for $i \in \mathcal{P}$ are not observed and cannot be used when trying to find weights solving Problem 2.1, and are only available for estimating the final estimate $\hat{\tau}_w$ *after* weights have been found.

## 2.2 COMMON METHODS IN WEIGHTING

In ATT/ATE estimation and transportability, true weights are proportional to the inverse of one of the propensity scores $p(A = a|X = x)$ [Ben-Michael et al., 2021] or $P(S = 1|X = x)$ [Cole and Stuart, 2010]. Thus, an inverse probability weighting estimator $\widehat{w}$ of $\frac{dQ_X}{dP_X}$ is obtained by fitting a model for the indicated propensity score and inverting it, leading to potentially outsize errors due to misspecification [Zubizarreta, 2015]. An alternative used in the automatic debiased machine learning (AutoDML) literature is to minimize the mean squared error between $\frac{dQ_X}{dP_X}$ and $\widehat{w}$, which can actually be estimated without exactly knowing the true weights $\frac{dQ_X}{dP_X}$ [Chernozhukov et al., 2022b,a, Newey and Newey, 2023]. Another family of methods [Hainmueller, 2012, Fong et al., 2018] relies on imposing that weights $w$ verify **balance** in some moments $r$, i.e. $\mathbb{E}_{P^w}[r(X)] = \mathbb{E}_Q[r(X)]$. Then one minimizes some dispersion measure of weights under these constraints. However, balancing $r(X)$ does not guarantee balancing the unknown $\mathbb{E}_P[Y|X]$ and the solution might not be feasible if $r$ has too many moments [Wainstein, 2022]. Similar methods enforce such balance approximately through a generalized method of moments [Imai and Ratkovic, 2014, Fong et al., 2018].

Finally, another family of methods [Ben-Michael et al., 2021] aims at finding weights $w$ minimizing $|\text{Bias}_{P,Q}(w)|$ where we refer to

$$\text{Bias}_{P,Q}(w) = \mathbb{E}_{P^w}[\mathbb{E}_P[Y|X]] - \mathbb{E}_Q[\mathbb{E}_P[Y|X]]$$

as the **"bias"** of weights $w$, measuring how short they fall of solving Problem 2.1 and which is also equal to the bias of the estimator $\hat{\tau}_w$ wrt the target estimand. It is usually assumed that $\mathbb{E}_P[\tilde{Y}|x]$ belongs to a class of functions $\mathcal{M}$ which leads to the bound

$$\begin{aligned}|\text{Bias}_{P,Q}(w)| &\leq \text{IPM}_{\mathcal{M}}(P_X^w, Q_X) \\ &:= \sup_{\bar{m} \in \mathcal{M}} |\mathbb{E}_{P^w}[\bar{m}(X)] - \mathbb{E}_Q[\bar{m}(X)]|\end{aligned}$$

where the RHS is an integral probability metric (IPM) [Spiperumbudur et al., 2012] on the class $\mathcal{M}$ and generally corresponds to a known probability discrepancy; for example the Wasserstein distance when $\mathcal{M}$ is the set of Lipschitz functions or the maximal mean discrepancy (MMD) wrt kernel $k$ when $\mathcal{M}$ is the RKHS of $k$. Thus, adding a term to control the variance of the weighting estimator [Kallus, 2020b, Ben-Michael et al., 2021], we obtain a solution $w$ by solving

$$\min_{w} \quad \text{IPM}_{\mathcal{M}}(P_X^w, Q_X)^2 + \sigma^2 \cdot ||w(X)||_{L_2(P)}^2 \quad (1)$$

for a chosen $\sigma > 0$ that controls a bias-variance trade-off [Bruns-Smith and Feller, 2022]. A key challenge is that as we do not know the outcome model $\mathbb{E}_P[\tilde{Y}|x]$, we do not know the model class $\mathcal{M}$, thus an adequate probability discrepancy to minimize. In practice, one resorts to trying a specific discrepancy, thus making an implicit assumption on the function space $\mathcal{M}$ which can then be inadequate wrt the outcome model $\mathbb{E}_P[\tilde{Y}|x]$ at stake. Recognizing this, directions in the literature include finding a data-driven tailored function class $\mathcal{M}$ [Kallus, 2020a, Wainstein, 2022] or finding guarantees when the function class is misspecified [Bruns-Smith and Feller, 2022].

## 2.3 CHOOSING A DISTANCE VIA A REPRESENTATION

Many methods minimize a probability discrepancy measure or more generally find weights that only depend on covariates $x$ via a vector-valued function $\phi(x)$ known as a **representation** [Kallus, 2020a, Xue et al., 2023]. Indeed, assuming any function class $\mathcal{M}$ implicitly assumes that any function linearly depends on a representation $\phi(x)$, e.g. the first-order moment $x$ for linear functions, the kernel feature spaces $k(., x)$ for the RKHS of kernel $k$ [Hazlett, 2020, Kallus, 2020a], and more generally $(m(x))_{m \in \mathcal{M}}$ for any class $\mathcal{M}$ (note that such a representation is not unique). In turn, every representation defines a function class. Thus, choosing a function class $\mathcal{M}$ means *implicitly* choosing a representation $\phi(x)$ and assuming that the true outcome model $\mathbb{E}_P[\tilde{Y}|x]$ linearly depends on it.

Further, it is also common practice to *explicitly* define a representation $\phi(x)$ (on which the outcome model need *not* depend linearly) and apply a weighting method using it. Notable examples include propensity scores and balancing scores [Rosenbaum and Rubin, 1983b], prognostic scores [Hansen, 2008] or variable selection [Brookhart et al., 2006, Colnet et al., 2024]. One motivation to do so is that a low-dimensional representation can mitigate undesirable effects of high dimensions in causal inference [Ning et al., 2020, D'Amour et al., 2021] or probability distances [Dudley, 1969, Ramdas et al., 2015] and improve efficiency by selecting essential covariate information wrt the DGP.

The question then becomes how to obtain suitable repre-

sentations $\phi(x)$. It is well-known that weighting on the true outcome model, the propensity score or a representation predicting either [Rosenbaum and Rubin, 1983b, Hansen, 2008] is a sensible choice as these representations preserve unconfoundedness. However, we do not have access to these true models or representations predicting them. Methods based on sufficient dimension reduction attempt to find a linear representation under the constraint that it predicts either model [Cook, 2009, Luo and Zhu, 2020], while others extract representations from a learnt model for the outcome, the treatment or the RCT indicator [Rosenbaum and Rubin, 1983a, Hansen, 2008, Cole and Stuart, 2010]. However, to the best of our knowledge, there are no guarantees on the bias when any posited model is misspecified or more generally when any underlying assumption is violated, while they are critical as one cannot verify such assumptions. In particular, classification-based learning of propensity scores does not optimize for covariate balance but for prediction of the treatment or the RCT indicator, while (near-)deterministic prediction of either will violate (strict [D'Amour et al., 2021]) overlap, leading to poor matching or weighting performance in practice [Alam et al., 2019, King and Nielsen, 2019]. In addition, many such methods learn the representation using outcomes, which is done before weighting, thus is not permitted in an actual design-based setting. More recent works learn implicit representations by positing a rich parametric class $\mathcal{M}$ [Ozery-Flato et al., 2018, Kallus, 2020a], as a result bias can be controlled but only for outcome models belonging to this class.

Thus, one might wonder whether guarantees on the bias can be provided when using *any* representation $\phi$ and *any* class $\mathcal{M}$, without using outcome information and without relying on rigid well-specification assumptions. This is the main contribution of our paper, which we develop next.

## 3 THEORY AND METHOD

### 3.1 QUANTIFYING THE INFORMATION LOSS

Choosing a representation $\phi(X)$ introduces many trade-offs. At one extreme, oracle representations, such as balancing scores or prognostic scores, perfectly preserve unconfoundedness; that is, unconfoundedness given $\phi(X)$ implies unconfoundedness given $X$. These are largely unknown, however. At the other extreme, degenerate representations, such as a *constant* $\phi(X)$, will destroy all the information in the original $X$. We now characterize representations that minimize the information lost relative to $X$.

Indeed, we first make technical assumptions ensuring that all expectations are well-defined. For any distribution $R$, random variable $Z$ and integer $p \geq 1$, let

$$||Z||_{L_p(R)} := (E_R[|Z|^p])^{\frac{1}{p}},$$

and note $Z \in L_P(R)$ iff $||Z||_{L_P(R)} < \infty$. Notably, for a

measurable function $f$ of values of $Z$,

$$||f||_{L_p(R_Z)} = (E_R[|f(Z)|^p])^{\frac{1}{p}} = ||f(Z)||_{L_p(R)}$$

We then make the following assumptions.

**Assumption 3.1.** $\frac{dQ_X}{dP_X}(X) \in L_2(P)$

**Assumption 3.2.** $\tilde{Y} \in L_2(P)$

Then, under Assumptions 2.3, 3.1, 3.2 by noting that for any weights $w$ wrt $P$ that are in $L_2(P_X)$, and for any measurable mapping $\phi(x)$ of covariates, the bias can be decomposed as

$$
\begin{aligned}
\text{Bias}_{P,Q}(w) &= \mathbb{E}_{P^w}\left[\mathbb{E}_P[\tilde{Y}|X]\right] - \mathbb{E}_Q\left[\mathbb{E}_P[\tilde{Y}|X]\right] \\
&= \underbrace{\mathbb{E}_{P^w}\left[\mathbb{E}_P[\tilde{Y}|\phi(X)]\right] - \mathbb{E}_Q\left[\mathbb{E}_P[\tilde{Y}|\phi(X)]\right]}_{\text{Bias wrt the representation}} \\
&\quad + \underbrace{\mathbb{E}_{P^w}\left[\mathbb{E}_P[\tilde{Y}|X] - \mathbb{E}_P[\tilde{Y}|\phi(X)]\right]}_{\text{Chosen weights bias}} \\
&\quad + \underbrace{\mathbb{E}_Q\left[\mathbb{E}_P[\tilde{Y}|\phi(X)] - \mathbb{E}_P[\tilde{Y}|X]\right]}_{\text{Confounding bias}}.
\end{aligned}
\quad (2)
$$

We now explain each term in the RHS. First, if the weights $w(X)$ are a function of the representation $\phi(X)$, the *bias wrt the representation* would be the bias if we replaced $X$ with $\phi(X)$ in the equality of Problem 2.1. This interpretation still holds for general weights $w(X)$ as from the tower property applied to $\mathbb{E}_{P^w}[\mathbb{E}[\tilde{Y}|\phi(X)]]$, they can be replaced with $\mathbb{E}_P[w(X)|\phi(X)]$, which is a $L_2(P_X)$ weight function wrt $P$ and is a function of $\phi(X)$, in the term. As in Section 2.2, we can directly bound the bias wrt the representation via an IPM of the form

$$\text{IPM}_{\mathcal{G}}(P^w_{\phi(X)}, Q_{\phi(X)}),$$

where for example, for a class $\mathcal{M}$ such that $\mathbb{E}_P[\tilde{Y}|x] \in \mathcal{M}$, the class $\mathcal{G}$ can contain

$$\phi(\mathcal{M}, P) := \{z \mapsto \mathbb{E}_P[m(X)|\phi(X) = z], \ m \in \mathcal{M}\}. \quad (3)$$

Second, the *chosen weights bias* measures how much "chosen" weights $w(x)$ do not depend on $\phi(x)$. It turns out that this quantity is zero for weights $\hat{w}(x)$ that solve the canonical minimization in Equation 1 with the aforementioned IPM; as we show next, these weights only depend on $\phi(x)$.

**Proposition 3.3.** *Let $\phi(x)$ be a measurable mapping with values in a space $\Phi$.*

1. *Under Assumptions 2.3, 3.1, if $\mathcal{G}$ is a class of $L_2(P_{\phi(X)})$ functions on $\Phi$, $\sigma > 0$, there is a unique solution $\hat{w}(x)$ to the problem*

$$\min_{\substack{w \text{ weight} \\ \text{function} \\ \text{wrt } P}} \text{IPM}_{\mathcal{G}}(P^w_{\phi(X)}, Q_{\phi(X)})^2 + \sigma^2 \cdot ||w(X)||^2_{L_2(P)}$$

*and it is a function of $\phi(x)$ $P_X$-almost surely, i.e. there exists $\bar{w} : \Phi \to \mathbb{R}$ such that $\hat{w}(x) = \bar{w}(\phi(x)) \, \forall x \, P_X$−a.s. ; and $\hat{w}(X) \in L_2(P)$.*

2. *Under Assumption 3.2, for any $L_2(P_X)$ weight function $w(x)$ wrt $P$ that is a function of $\phi(x)$ $P_X$-a.s., the chosen weights bias is zero.*

Finally, the **confounding bias** is the most important term of this decomposition, as it characterizes the information lost in $\phi(X)$ relative to $X$ — and thus can be seen as the bias *of* $\phi$, rather than the bias *wrt* $\phi$ that is applied to weights.

When the target is $\mathbb{E}[Y(a)]$, this quantity is the difference between $\mathbb{E}\left[\mathbb{E}[Y \mid A = a, \phi(X)]\right]$ and $\mathbb{E}\left[\mathbb{E}[Y \mid A = a, X]\right]$, measuring how much $\phi(X)$ preserves unconfoundedness [D'Amour and Franks, 2021, Melnychuk et al., 2024]. More generally, for solution weights $\hat{w}$ of Equation 1 with an IPM depending on $\phi(X)$, it is exactly the difference between the biases of $\hat{w}$ wrt original covariates $X$ and their representation $\phi(X)$, as shown by Equation 2. Thus, if $\hat{w}$ has a small (resp. zero) bias wrt $\phi$, then it will also have a small (resp. zero) bias overall.

To the best of our knowledge, this is the first extension of the confounding bias for the $\mathbb{E}[Y(a)]$ target estimand to more general weighting problems in causal inference. It has a similar formulae as the *excess target information loss* in Johansson et al. [2019] measuring the loss of information induced by a representation in domain adaptation. We further provide a characterisation for it that will prove useful.

**Proposition 3.4.** *Under Assumption 2.3, for any measurable $\phi(x)$, $Q_{\phi(X)}$ is absolutely continuous wrt $P_{\phi(X)}$, with*

$$\frac{dQ_{\phi(X)}}{dP_{\phi(X)}}(\phi(X)) = \mathbb{E}_P\left[\frac{dQ_X}{dP_X}(X)\Big|\phi(X)\right] \quad P\text{-a.s.}$$

*and under the additional Assumptions 3.1 and 3.2, the confounding bias is equal to both*

$$\mathbb{E}_P\Bigg[\left(\mathbb{E}_P[\tilde{Y}|\phi(X)] - \mathbb{E}_P[\tilde{Y}|X]\right)$$
$$\times \left(\frac{dQ_X}{dP_X}(X) - \frac{dQ_{\phi(X)}}{dP_{\phi(X)}}(\phi(X))\right)\Bigg] \quad (4)$$

*and*

$$-\mathbb{E}_P\left[\mathbb{E}_P[\tilde{Y}|X]\left(\frac{dQ_X}{dP_X}(X) - \frac{dQ_{\phi(X)}}{dP_{\phi(X)}}(\phi(X))\right)\right] \quad (5)$$

When the confounding bias is zero, $\phi$ is known as a *deconfounding score* [D'Amour and Franks, 2021], and the overall bias is simply equal to the bias wrt $\phi$. In particular, from Equation 4, the confounding bias will be zero in two special cases :

• When $\mathbb{E}_P[\tilde{Y}|X] = \mathbb{E}_P[\tilde{Y}|\phi(X)]$ $P$−a.s., that is

$$\mathbb{E}_P[\tilde{Y}|X] = \mathbb{E}_P\left[\mathbb{E}_P[\tilde{Y}|X]\Big|\phi(X)\right] \quad P\text{-a.s.}$$

from the tower property. This is equivalent to $\mathbb{E}_P[\tilde{Y}|x]$ being a function of $\phi(x)$ $P_X$-a.s., i.e. $\phi(X)$ a prognostic score [Hansen, 2008].

• When $\frac{dQ_X}{dP_X}(X) = \frac{dQ_{\phi(X)}}{dP_{\phi(X)}}(\phi(X))$ $P$-a.s., that is

$$\frac{dQ_X}{dP_X}(X) = \mathbb{E}_P\left[\frac{dQ_X}{dP_X}(X)\Big|\phi(X)\right] \quad P\text{-a.s.}$$

from Proposition 3.4. This is equivalent to $\frac{dQ_X}{dP_X}(x)$ being a function of $\phi(x)$ $P_X$-a.s., i.e. $\phi(X)$ a balancing score [Rosenbaum and Rubin, 1983b].

We make a more rigorous connection between the confounding bias and canonical scores from the literature as well as notions from transportability in Appendix B

Further, the confounding bias and its role in the decomposition of Equation 2 allow us to extend the idea of a deconfounding score to hold approximately, rather than exactly. Indeed, if the confounding bias of $\phi$ is not zero but remains small, then we can expect that a small bias wrt $\phi$ obtained by solving the problem of Proposition 3.3 will still yield a small overall bias. This gives us more flexibility than relying on well-specified models, where any guarantee on the bias is lost in case of misspecification. In contrast, the confounding bias directly quantifies the misspecification itself.

Thus, one might wonder whether we can minimize directly said misspecification to find an *approximate* deconfounding score $\phi$. However Equation 4 involves ground-truth models we do not have access to like $\mathbb{E}_P[\tilde{Y}|x]$, $\frac{dQ_X}{dP_X}(x)$ as well as their projections on $\phi(x)$. Further, we do not observe any outcomes at this stage, precluding any estimation of $\mathbb{E}_P[\tilde{Y}|x]$. To address all of this, note that a direct application of the Cauchy-Schwarz inequality to Equation 5 yields

$$|\text{Confounding bias}| \leq ||\mathbb{E}_P[\tilde{Y}|X]||_{L^2(P)} \cdot \text{BSE}_{P,Q}(\phi) \quad (6)$$

where we further have $||\mathbb{E}_P[\tilde{Y}|X]||_{L^2(P)} \leq ||\tilde{Y}||_{L_2(P)}$ from Jensen's inequality, and we call

$$\text{BSE}_{P,Q}(\phi) := \left\|\frac{dQ_X}{dP_X}(X) - \frac{dQ_{\phi(X)}}{dP_{\phi(X)}}(\phi(X))\right\|_{L_2(P)} \quad (7)$$

the **balancing score error** (BSE). This name is justified as from Proposition 3.4, this quantity is equal to

$$\left\|\frac{dQ_X}{dP_X}(X) - \mathbb{E}_P\left[\frac{dQ_X}{dP_X}(X)\Big|\phi(X)\right]\right\|_{L_2(P)},$$

that is the root mean-squared error between $\frac{dQ_X}{dP_X}(X)$ and its projection on $\phi(X)$, i.e. its best predictor from $\phi(X)$ in $L_2(P)$. In other words, it measures the extent to which $\frac{dQ_X}{dP_X}(x)$ is not a function of $\phi(x)$ $P_X$-a.s., and therefore the extent to which $\phi(x)$ is not a balancing score. Importantly, it does not depend on the pseudo-outcome $\tilde{Y}$, only

on the marginal $P_X$. Note that the confounding bias can be zero and the balancing score error positive, even potentially arbitrary, for many representations $\phi(x)$ that contain information on the outcome model $\mathbb{E}_P[\tilde{Y}|x]$. Concrete examples include prognostic scores from Hansen [2008], or the deconfounding scores in the example of Section 5 in D'Amour and Franks [2021]. Our setup excludes such representations as it assumes that we do not observe outcomes at this stage. Alternatively, if one had access to outcomes, then similarly as for the balancing score error, we can bound the confounding bias with a "prognostic score error".

On the other hand, note that the balancing score error allows us to control the resulting bias with only mild assumptions on the outcome model. We formalize this next.

**Proposition 3.5.** *Under Assumptions 2.3, 3.1, 3.2, for any set $\mathcal{M}$ of $L_2(P_X)$ functions such that $E_P[\tilde{Y}|x] \in \mathcal{M}$, for any measurable representation $\phi$, and for any $L_2(P_X)$ weights $w$ wrt $P$ depending on $\phi(x)$ $P_X$-a.s., defining $\phi(\mathcal{M}, P)$ as in Equation 3,*

$$|Bias_{P,Q}(w)| \leq IPM_{\phi(\mathcal{M},P)}(P^w_{\phi(X)}, Q_{\phi(X)})$$
$$+ ||\tilde{Y}||_{L_2(P)} \cdot BSE_{P,Q}(\phi).$$

We note that the bound of Proposition 3.5 is "sharp" in the sense that when we replace the IPM and the BSE by the (unknown) terms they bound, namely the bias wrt the representation and the confounding bias, the inequality becomes an equality. It further suggests a two-step approach to minimize the overall bias on the LHS. First, learn a representation $\phi$ that minimizes the BSE, i.e. the second term of the RHS, plug this learnt representation $\phi$ into an IPM and find weights minimizing it, or in other words minimizing the first term of the RHS. To learn the representation, the BSE could be used in addition or in replacement of the traditional likelihood to learn propensity score models. Importantly, we can bypass propensity score estimation altogether and posit more general representations, including multivariate functions. We turn to this in the next sections.

### 3.2 OPERATIONALIZING AND MINIMIZING INFORMATION LOSS

While we have avoided the need to specify an outcome model $\mathbb{E}_P[\tilde{Y}|x]$, a key bottleneck remains for the balancing score error: we do not have access to the true weights $\frac{dQ_X}{dP_X}(X)$ or their projection $\frac{dQ_{\phi(X)}}{dP_{\phi(X)}}(\phi(X))$. One possible workaround is to first remove the projection by using the definition of a conditional expectation: for any function $g$ on the image space of $\phi$,

$$\text{BSE}_{P,Q}(\phi) \leq \left|\left|\frac{dQ_X}{dP_X}(X) - g(\phi(X))\right|\right|_{L^2(P)}. \quad (8)$$

In particular, for $\epsilon > 0$, if there exists *any* function $g$ on the image space of $\phi$ such that the RHS of Equation 8 is

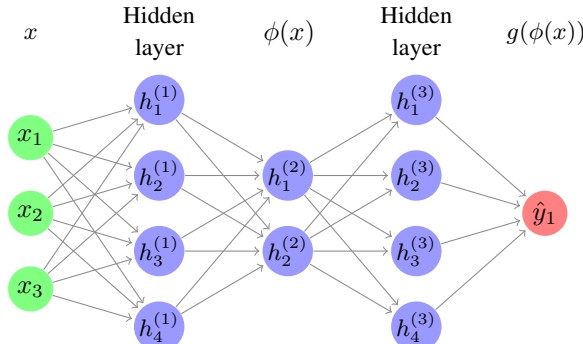

Figure 1: Neural Network to Learn a Representation $\phi$.

below $\epsilon/||\tilde{Y}||_{L_2(P)}$, then $\phi$ has an absolute confounding bias at most $\epsilon$. This gives us more flexibility than working with the true projection of $\frac{dQ_X}{dP_X}$, and motivates finding an $g$ and $\phi$ minimizing the RHS.

This approach, however, is insufficient since we still do not have access to $\frac{dQ_X}{dP_X}$. A key result from the covariate shift literature [Kanamori et al., 2009], notably exploited in the AutoDML literature [Chernozhukov et al., 2022a,b], helps us remove $\frac{dQ_X}{dP_X}$ from the minimization entirely : for any distributions $P, Q$ verifying Assumption 2.3, and for any function $v$, $\left|\left|\frac{dQ_X}{dP_X}(X) - v(X)\right|\right|^2_{L^2(P)}$ is equal to $\mathcal{L}_{P,Q}(v)$ up to an additive constant wrt $v$, where we refer to

$$\mathcal{L}_{P,Q}(v) := \mathbb{E}_P[v(X)^2] - 2 \cdot \mathbb{E}_Q[v(X)]$$

as the *AutoDML loss*. In particular, $\mathcal{L}_{P,Q}(v)$ can be estimated in finite samples for any known $v$, as

$$\mathcal{L}_{\mathcal{P},\mathcal{Q}}(v) = \frac{1}{|\mathcal{P}|}\sum_{i \in \mathcal{P}} v(X_i)^2 - \frac{2}{|\mathcal{Q}|}\sum_{i \in \mathcal{Q}} v(X_i)$$

This motivates an approach to *learn a representation $\phi$*. We posit a parameterized representation $\phi(x; \theta_\phi)$ with values in a space $\Phi$, and a scalar parameterized function $g(.; \theta_g)$ on $\Phi$. Then we minimize $\mathcal{L}_{\mathcal{P},\mathcal{Q}}(g(\phi(.; \theta_\phi); \theta_g))$ wrt $\theta_\phi, \theta_g$. Due to the compositionality of neural networks, we parameterize $g$ and $\phi$ jointly in a neural network which is plugged into the AutoDML loss, similarly to the Riesz representer component of RieszNet [Chernozhukov et al., 2022a], and where a pre-specified, potentially low-dimensional hidden layer is later used as the representation $\phi$ [Clivio et al., 2022]. This is illustrated in Figure 1. Unlike RieszNet, we do not use any outcome information and we do not use the final Riesz representer head as the solution weight function, but instead plug the representation into a probability distance to obtain such a solution, as we will see shortly. We later show that this yields lower biases in our experiments.

## 3.3 EXTENSION TO SIMULTANEOUS WEIGHTINGS

In ATE estimation, one aims at estimating all $\mu(a) := \mathbb{E}[Y(a)]$ for all $a \in \mathcal{A}$ simultaneously. This can be done [Martinet, 2020] by finding a function $f(a)$ minimizing

$$\mathbb{E}[(\mu(A) - f(A))^2]$$

over functions $f$ defined by

$$f(a) = \mathbb{E}[w_a(X)\mathbb{E}[Y|X, A = a] \mid A = a]$$

where $w_a(X)$ is a weight function wrt $P_{X|A=a}^{\text{data}}$. This is equivalent to minimizing

$$\mathbb{E}[\text{Bias}^2_{P^{\text{data}}(.|A), P^{\text{data}}(.)}(w_A)],$$

which is a special case of minimizing the **joint squared bias**

$$\text{Bias}^2_{P^\Lambda, Q^\Lambda, p_\Lambda}(w^\Lambda) := \mathbb{E}_{p_\Lambda(\alpha)}[\text{Bias}^2_{P^\alpha, Q^\alpha}(w^\alpha)]$$

where $\alpha$ belongs to a set $\Lambda$ endowed with a probability distribution $p_\Lambda(\alpha)$, $h^\Lambda := (h^\alpha)_{\alpha \in \Lambda}$ for any $h$, and $P^\alpha, Q^\alpha, w^\alpha$ are a source distribution, a target distribution, a weight function indexed by $\alpha \in \Lambda$, respectively. The following corollary extends previous results on the balancing score error to the setting of simultaneous weighting problems.

**Corollary 3.6.** *Let $\Lambda$ be a set endowed with a distribution $p_\Lambda(\alpha)$, $P^\Lambda, Q^\Lambda$ be mappings from $\Lambda$ to a distribution such that for any $\alpha \in \Lambda$, $P^\alpha, Q^\alpha$ satisfy Assumptions 2.3, 3.1, 3.2. Then for any $\mathcal{M}^\Lambda$ such that $\forall \alpha \in \Lambda$, $\mathbb{E}_{P^\alpha}[\tilde{Y}|x] \in \mathcal{M}^\alpha$ where $\mathcal{M}^\alpha$ is a set of $L_2(P_X)$ functions, for any mapping $\phi^\Lambda$ from $\Lambda$ to measurable representations, for any $w^\Lambda$ such that each $w^\alpha(x)$ is an $L_2(P_X^\alpha)$ weight function wrt $P^\alpha$ depending on $\phi^\alpha(x)$,*

$$\frac{1}{2} \cdot Bias^2_{P^\Lambda, Q^\Lambda}(w^\Lambda)$$

$$\leq \mathbb{E}_{p_\Lambda(\alpha)}\left[IPM^2_{\phi^\alpha(\mathcal{M}^\alpha, P^\alpha)}(P^{\alpha, w^\alpha}_{\phi^\alpha(X)}, Q^\alpha_{\phi^\alpha(X)})\right]$$

$$+ \left(\sup_{\alpha \in \Lambda} ||\tilde{Y}||^2_{L_2(P^\alpha)}\right) \cdot BSE^2_{P^\Lambda, Q^\Lambda, p_\Lambda}(\phi^\Lambda).$$

*where we call*

$$BSE^2_{P^\Lambda, Q^\Lambda, p_\Lambda}(\phi^\Lambda) := \mathbb{E}_{p_\Lambda(\alpha)}[BSE^2_{P^\alpha, Q^\alpha}(\phi^\alpha)]$$

*the **joint squared balancing score error**.*

We also note that this framework is identical to Problem 2.1 when $\Lambda$ is of cardinality 1. Finally, we can extend the previous section to simultaneous weights, where we now find an indexed representation $\phi^\Lambda$ that minimizes the joint squared balancing score error. We do so by first positing a parameterized representation $\phi(x, \alpha; \theta_\phi)$ belonging to some

space $\Phi$ and a scalar parameterized function $g(\varphi, \alpha; \theta_g)$ on the $\Phi \times \Lambda$ space, and then minimizing

$$\mathcal{L}^{g, \phi, p_\Lambda}_{\mathcal{P}^\Lambda, \mathcal{Q}^\Lambda}(\theta) = \mathbb{E}_{p_\Lambda(\alpha)}\left[\mathcal{L}_{\mathcal{P}^\alpha, \mathcal{Q}^\alpha}(g(\phi(., \alpha; \theta_\phi), \alpha; \theta_g))\right]$$

wrt $\theta_\phi, \theta_g$, where $\mathcal{P}^\alpha, \mathcal{Q}^\alpha$ are samples from $P^\alpha, Q^\alpha$.

If desired, we can separate the problem of minimizing the joint squared bias into independent weighting problems, minimizing each individual bias separately, especially when $\Lambda$ is finite and with few elements. However, we can also share parameters or dependencies between individual problems, e.g. use the same representation for all problems, i.e. $\phi^\alpha := \phi$ for some $\phi$ for all $\alpha \in \Lambda$, or share parameters $\theta_\phi, \theta_g$ between problems $\alpha \in \Lambda$, notably when there are few samples for every $\mathcal{P}_\alpha$ or $\mathcal{Q}_\alpha$ as in ATE estimation with high-cardinal $\mathcal{A}$.

For completeness, we now give examples of $\Lambda, \mathcal{P}^\Lambda, \mathcal{Q}^\Lambda$. In ATE estimation, we have access to samples $\{(x_i, a_i, y_i)\}_{i=1,\cdots,n}$ of $P^{\text{data}}(X, A, Y)$. Then, $\Lambda = \mathcal{A}$ and for each $\alpha = a \in \mathcal{A}$, $\mathcal{P}^a = \{(x_i, y_i)\}_{i:a_i=a}$, $\mathcal{Q}^a = \mathcal{Q}^0 := \{x_i\}_{i=1,\cdots,n}$. In ATT estimation, where $\mathcal{A}$ is binary, then $\Lambda = \{0\}$, $\mathcal{P}^0 = \{(x_i, y_i)\}_{i:a_i=0}$, $\mathcal{Q}^0 = \{x_i\}_{i:a_i=1}$. In transportability, $\Lambda = \{0\}$, we have access to samples $\{(x_i, a_i, y_i)\}_{i=1,\cdots,n}$ of the RCT distribution $P^{\text{data}}(X, A, Y|S = 1)$, samples $\{(x_i)\}_{i=n+1,\cdots,n+m}$ of some observational data $P(X|S = 0)$, and $\pi = P^{\text{data}}(A = 1|S = 1)$, so $\Lambda = \{0\}$, $\mathcal{P}^0 = \{(x_i, \tilde{y}_i = \frac{a_i y_i}{\pi} - \frac{(1-a_i)y_i}{1-\pi})\}_{i=1,\cdots,n}$, $\mathcal{Q}^0 = \{x_i\}_{i=n+1,\cdots,n+m}$.

## 3.4 WEIGHTING AND ALGORITHM

Learning a representation by minimizing a bound of the BSE helped us minimize the second term of the RHS of Proposition 3.5. We now turn to minimizing the first term, that is **finding weights**. In finite samples, we aim to find discrete weights $w_i = w(X_i)$ for $i \in \mathcal{P}$, with constraints $\forall i \in \mathcal{P}$, $w_i \geq 0$ and $\frac{1}{|\mathcal{P}|}\sum_{i \in \mathcal{P}} w_i = 1$.

In line with Proposition 3.3, we would ideally obtain $\hat{w}$ by solving Equation 1 with $IPM_{\phi(\mathcal{M}, \mathcal{P})}(\mathcal{P}^w_{\phi(X)}, \mathcal{Q}_{\phi(X)})$ where $\mathcal{P}_w$ is the empirical distribution over $\mathcal{P}$ with probabilities $w_i/|\mathcal{P}|$. However, as $\mathcal{M}$ is unknown, $IPM_{\phi(\mathcal{M}, P)}$ will remain unknown. Proposition 9 of Clivio et al. [2022] suggests that if $\mathcal{M}$ is the set of $L$-Lipschitz constants and $\phi(x)$ is a neural network with invertible and bi-Lipschitz activation functions, then $\phi(\mathcal{M}, P)$ is contained in the class of $L'$-Lipschitz functions for some $L'$ that depends on the weights and bi-Lipschitz constant of $\phi$ and might be significantly larger than $L$.

For computational simplicity, we work with a canonical IPM and choose the maximal mean discrepancy wrt some kernel $k$ [Gretton et al., 2012], following common practice in the literature [Kallus, 2020b, Huling and Mak, 2024].

---

**Input :** Distribution $p_\Lambda(\alpha)$ over $\alpha \in \Lambda$, model
$g(\phi(.,\alpha;\theta_\phi),\alpha;\theta_g)$, for each $\alpha$: samples $\mathcal{P}^\alpha$, $\mathcal{Q}^\alpha$,
kernel $k^\alpha$, hyperparameter $\sigma^\alpha \geq 0$.

---

Initialize $\theta := (\theta_\phi, \theta_g)$;
**while** $\theta$ *not converged* **do**
    Move $\theta$ in direction $-\nabla_\theta \mathcal{L}^{g,\phi,p}_{\mathcal{P}^\Lambda,\mathcal{Q}^\Lambda}(\theta)$;
**end**
**for** $\alpha \in \Lambda$ **do**
    $\phi^\alpha(x) \leftarrow \phi(x, \alpha, \theta_\phi)$;
    $\tilde{k}^\alpha(x, x') \leftarrow k^\alpha(\phi^\alpha(x), \phi^\alpha(x'))$;
    $\hat{w}^\alpha \leftarrow$ kernel optimal matching with simplex
      weights, kernel $\tilde{k}^\alpha$ and regularization
      hyperparameter $(\sigma^\alpha)^2$ ;
**end**
**Result:** $\hat{w}^\Lambda$

---

**Algorithm 1:** Representation Learning and Weighting.

More generally, minimizing such an MMD under the above weight constraints is referred to as *kernel optimal matching* (KOM) with simplex weights [Kallus, 2020b] in causal inference, where we change the setting from treated and control distributions to source and control distributions, or empirical *kernel mean matching* (KMM) [Huang et al., 2006] in covariate shift, where we add $L_2$ regularization. This minimization amounts to solving a quadratic program (QP) with linear constraints, which can be done using any off-the-shelf QP solver. The $\sigma^2$ hyperparameter for regularization can be selected either with a fixed value (e.g. 0 as in Huling and Mak [2024]) or from a principled procedure [Kallus, 2020b]. In the case of simultaneous weightings, this procedure can be repeated for each problem indexed $\alpha = 1, \cdots, \ell$. Our exact implementation of kernel optimal/mean matching for this purpose is given in Appendix E

We summarize all the previous steps in Algorithm 1. Each component $\hat{w}^\alpha$ of its result $\hat{w}^\Lambda$ is then plugged in an estimator $\hat{\tau}^\alpha_{\hat{w}^\alpha}$ of $\mathbb{E}_{\mathcal{P}^\alpha_{\hat{w}^\alpha}}[\mathbb{E}_{P^\alpha}[\tilde{Y}|X]]$ as

$$\hat{\tau}^\alpha_{\hat{w}^\alpha} = \frac{1}{|\mathcal{P}^\alpha|} \sum_{i \in \mathcal{P}^\alpha} \hat{w}^\alpha_i \tilde{Y}_i.$$

This estimator could be analyzed theoretically (e.g. for consistency, error rates, ...) by inspecting, for each $\alpha \in \Lambda$, two separate terms : (i) the confounding bias of the learnt representation $\phi^\alpha$, and (ii) the difference between the estimator and the representation-wise estimand $\mathbb{E}_{Q^\alpha}[\mathbb{E}_{P^\alpha}[\tilde{Y}|\hat{\phi}^\alpha(X)]]$. As the representation is learnt using the same loss as Equation 2.6 of Chernozhukov et al. [2024] and the confounding bias of the learnt representation is bounded by its balancing score error, itself bounded by the Riesz representer error in Theorem 2.1 of Chernozhukov et al. [2024], we can resort to their results. Then, the difference between estimator and representation-wise estimand can be analyzed using previ-

ous work on analysis of KOM or KMM, such as Kallus [2020b] or Yu and Szepesvári [2012].

## 4 RELATED WORK

**Generalization bounds.** An adjacent line of work to ours is generalization bounds in domain adaptation, where one aims to bound the risk of a model on a target domain using the risk on a source domain. Typically, this involves a representation and the bound includes an IPM analogous to ours [Ben-David et al., 2006, Zhao et al., 2018a,b, Li et al., 2023]. In extensions of such bounds to causal inference, where a counterfactual risk is bounded using a factual risk and an IPM as before but the representation is usually assumed to be invertible [Shalit et al., 2017, Bellot et al., 2022, Johansson et al., 2022, Kazemi and Ester, 2024], precluding the study of misspecified or confounded representations. Thus, usually no term quantifying the "misspecification" of the representation is added. Notable exceptions are Johansson et al. [2019] in domain generalization and Curth et al. [2021] in causal inference which include an *information loss* without actively trying to minimize it. The information loss from Johansson et al. [2019] can be shown to be identical to our confounding bias with the outcome replaced by the loss function. A balancing score error analogous to ours will bound this information loss if the loss function is bounded above by a constant and our AutoDML loss-based approach can be used too.

**Confounding bias, balancing score error** D'Amour and Franks [2021] also define a confounding bias and their Proposition 2 can be shown to be a special case of our Proposition 3.4 for ATE estimation, which they only compute in a restricted case with Gaussian covariates and generalized linear models for outcome and propensity models. Melnychuk et al. [2024] define a conditional confounding bias and estimate bounds of it for a *fixed* representation instead of learning it using their bounds, which does not seem trivial as their estimation relies on two different neural network fitting steps *after* fitting the representation. Clivio et al. [2022] provide an alternative error on how much the representation is not a balancing score but they mention that it is difficult to compute and do not use it to learn the representation, which relies on assuming a propensity score model. Further, note that approaches to sensitivity analysis generally derive or bound the confounding bias induced by not including unobserved confounders in the adjustment set [Imbens, 2003, Tan, 2006, Hartman and Huang, 2024], although this is done by making about assumptions on the relationship between unobserved confounders and other variables in the data generating process ; in contrast, aforementionned methods and our work pertain to a setting without such unobserved confounders.

**Learning representations for treatment effects.** For weighting, besides points developed in Section 2.3, Deep-

Match [Kallus, 2020a] requires a grid search involving multiple neural network trainings (50 in the experiments) and other models [Averitt et al., 2020, Kitazawa, 2022] take an $f$-divergence as the discrepancy measure but do not provide bounds on the bias, which is likely inherent to the non-intersection of IPMs and $f$-divergences [Sriperumbudur et al., 2012]. Other methods learn representations using outcome regression, alone or with weights [Shalit et al., 2017, Johansson et al., 2022, Chernozhukov et al., 2022a].

**Outcome-based weights and representations** Some methods use outcomes to derive (i) the outcome function class $\mathcal{M}$ e.g. as a confidence interval around a regressed outcome model as in Wainstein [2022]; (ii) the representation $\phi$ as in the canonical prognostic scores [Hansen, 2008] or more recent and more general deconfounding scores [D'Amour and Franks, 2021]; or (iii) the weights more generally, e.g. by directly estimating the density ratio between the source and target distributions of the outcome [Taufiq et al., 2023]. Finally, many standard outcome modeling approaches, such as (kernel ridge) regression are implicitly weighting estimators so one could use such approaches to derive weights; see, for example, Bruns-Smith et al. [2023].

## 5 NUMERICAL RESULTS

We now evaluate our method and alternatives on the IHDP [Hill, 2011] and News datasets [Johansson et al., 2016] for ATE estimation and a Traumatic Brain Injury (TBI) dataset [Colnet et al., 2024] for transportability.

For weighting, we focus on KOM for two kernels, 1) the *energy distance* kernel $k(x, x') = -||x - x'||_2$; KOM with this kernel is known as *energy balancing* [Huling and Mak, 2024] ; 2) the linear kernel $k(x, x') = x^T x'$. We evaluate these two methods with original covariates ("Energy" and "Linear"), a representation learned according to our approach ("Ours + Energy" and "Ours + Linear"), one through the canonical Principal Component Analysis (PCA, Hotelling [1933]) approach ("PCA + Energy" and "PCA + Linear"), the propensity score model vector $((\hat{p}(a|x))_{a \in \mathcal{A}}$ for ATE estimation, $(\hat{p}(s|x))_{s=0,1}$ for transportability) learnt with a gradient boosting classifier ("PS + Energy" and "PS + Linear"), representations from a layer of a neural network model of such propensity score models as in neural score matching (NSM, Clivio et al. [2022]) ("NSM + Energy" and "NSM + Linear"). We also check IPW with the same propensity scores ("IPW"), entropy balancing with first-order moments ("Entropy"), the weights head of the neural network used to train our representation("NN Head"), and uniform weights ("Unweighted"). Weights from "IPW" and "NN Head" were normalized to prevent outsize errors, while those from other methods were already normalized by design.

On energy balancing or linear kernel methods, we take

$\sigma^\alpha = 0.01$. KOM was performed using the `osqp` library in Python [Stellato et al., 2020], in line with the implementation of energy balancing in the `weightit` package [Greifer, 2024]. All representations are 10-dimensional, and we always use a common representation for all treatment arms. The neural network first has a 200-unit layer, a 10-unit layer corresponding to the representation, a second 200-unit layer, and finally the scalar head. Neural network implementation was performed in PyTorch [Paszke et al., 2019]. Adam [Kingma and Ba, 2015] was used to optimize the loss with a learning rate of $0.01$ and early stopping with a patience of 3 epochs, and all other hyperparameters at their default PyTorch values. We average results over 50 random seeds for IHDP and News, 100 for TBI. We show standard errors in parentheses.

Table 1: Joint Bias on the IHDP, News and TBI Datasets

| Method | IHDP | News | TBI |
|---|---|---|---|
| Ours + Energy | 0.079 (0.011) | 0.128 (0.014) | 5.00 (0.37) |
| NSM + Energy | 0.167 (0.040) | 0.070 (0.013) | 5.40 (0.53) |
| PS + Energy | 0.096 (0.012) | 0.381 (0.026) | 7.79 (0.53) |
| PCA + Energy | 0.080 (0.014) | 0.314 (0.020) | 10.65 (0.82) |
| Energy | 0.078 (0.014) | 0.397 (0.027) | 10.69 (0.83) |
| Ours + Linear | 0.087 (0.009) | 0.122 (0.013) | 18.50 (1.61) |
| NSM + Linear | 0.183 (0.043) | 0.113 (0.018) | 19.20 (1.71) |
| PS + Linear | 0.105 (0.017) | 0.499 (0.036) | 13.22 (1.15) |
| PCA + Linear | 0.077 (0.013) | 0.321 (0.023) | 63.86 (2.29) |
| Linear | 0.076 (0.011) | 0.168 (0.011) | 22.71 (1.75) |
| Entropy | 0.087 (0.013) | 0.221 (0.020) | 7.63 (0.60) |
| IPW | 0.114 (0.024) | 0.280 (0.018) | 2.28 (0.18) |
| NN Head | 0.181 (0.031) | 0.746 (0.121) | 59.71 (2.52) |
| Unweighted | 0.195 (0.050 | 0.611 (0.053) | 7.67 (0.15) |

As a metric, we consider the joint bias (JB), which is the square-root of the joint squared bias where the target esti-

mand is replaced by the average (known) outcome model over the empirical target distribution,

$$\sqrt{\sum_{\alpha \in \Lambda} p_\Lambda(\alpha) \left( \frac{\sum_{i \in \mathcal{P}^\alpha} w_i^\alpha \tilde{Y}_i}{|\mathcal{P}^\alpha|} - \frac{\sum_{i \in \mathcal{Q}^\alpha} \mathbb{E}_{\mathcal{P}^\alpha}[\tilde{Y}|x_i]}{|\mathcal{Q}^\alpha|} \right)^2}.$$

Results are shown in Table 1. For either energy or linear KOM, our representation typically outperforms all other representations; exceptions are NSM on News for both kernels, the propensity score on TBI with the linear kernel, original covariates for both kernels and PCA for the linear kernel on IHDP. It further outperforms baselines not relying on KOM on IHDP and News for both kernels and on TBI for the energy balancing kernel, except entropy balancing for the linear kernel on IHDP and IPW on News. We note that the linear kernel yields generally degraded performance on TBI compared to the energy balancing kernel, but not other datasets. On IHDP, each KOM method performs better using original covariates than using a representation, which suggests that dimensionality reduction in any form is not necessarily beneficial on such a dataset where 16 out of 25 covariates are binary. Notably, on all datasets, using our representation with any KOM outperforms the Riesz representer head of the same neural network used to train the representation. Further, on the 3477-dimensional News dataset, energy balancing was significantly sped up when using a lower-dimensional representation instead of original covariates.

On TBI, high biases are due to a wide range of pseudo-outcomes (e.g. from $-8.37$ to $174.18$, with a target estimand at $56.89$ on seed 5), and the highest biases to weights with most of their mass on a single point with an pseudo-outcome far away from the target estimand (e.g. $97\%$ of the mass on an pseudo-outcome of $145.40$ for NN Head, compared to at most $5\%$ on an pseudo-outcome of $117.97$ for entropy balancing, still on seed 5).

## 6 LIMITATIONS AND CONCLUSION

We have shown the importance of the confounding bias and the balancing score error (BSE) in learning representations for weighting, and have outlined a method to minimize the BSE. Experimental results suggest that representations obtained from the method might help improve performance for common optimization-based weighting approaches. The method could notably be applied to multimodal data involving tabular, text and image covariates [Klaassen et al., 2024].

One concern could be that the functions $g, \phi$ are generally not uniquely identifiable by minimizing the AutoDML loss. Without restrictions, many different $(g, \phi)$ tuples will indeed share the same value of the AutoDML loss, e.g. any $(g_h, \phi_h) = (g \circ h, h^{-1}(\phi))$ where $h$ is invertible. However, restricting $g$ and $\phi$ to be components of a neural network with a given architecture will exclude many possible invertible $h$'s. Some $h$'s will remain though, such as $h(z) = \lambda \odot z$ where $\odot$ is the Hadamard product and $\lambda_i \neq 0 \; \forall i$, which means that the returned $\phi$ might have arbitrary amplitude or smoothness. A workaround could be in adding some regularization of $\phi$ in the AutoDML loss, eg through weight decay. We do not perform weight decay and still obtain competitive performance in later experiments, which suggests that Adam optimization might choose an appropriate $\phi$ in practice.

Directions for future work to address limitations of our current approach include: (1) check whether such quantification of the representation's quality can also be done for augmented estimators, (2) evaluate the currently unknown gaps between the confounding bias and the BSE, and between the BSE and the AutoDML objective ; the latter provides a worst-case error but can be overly conservative ; (3) characterize the function class of the projection of the outcome model on the representation, depending on the class of the original outcome model or that of the representation, instead of assuming a canonical RKHS as we do now ; (4) develop a more thorough theoretical analysis of the estimator than the strategy presented in this paper.

**Acknowledgements**

We sincerely thank David Bruns-Smith, Sam Pimentel, Erin Hartman and anonymous reviewers for valuable feedback. O.C. was supported by the EPSRC Centre for Doctoral Training in Modern Statistics and Statistical Machine Learning (EP/S023151/1). A.F. was supported in part by the Institute of Education Sciences, U.S. Department of Education, through Grant R305D200010. C.H. was supported by the EPSRC Bayes4Health programme grant and The Alan Turing Institute, UK.

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

# Towards Representation Learning for Weighting Problems in Design-Based Causal Inference
# (Supplementary Material)

**Oscar Clivio**[1]                **Avi Feller**[2]                **Chris Holmes**[1]

[1]Department of Statistics, University of Oxford
[2]Goldman School of Public Policy and Department of Statistics, University of California, Berkeley

## A   DETAILS ON PROBLEMS IN CAUSAL INFERENCE

Under the assumptions of *no interference* and *consistency*, $A = a$ implies $Y = Y(a)$, which can written as $Y = \sum_{a \in \mathcal{A}} 1_{\{A=a\}} Y(a)$ or, more compactly, $Y = Y(A)$. Further, under *unconfoundedness* and *overlap* we have that $\mathbb{E}[Y(a)|X] = \mathbb{E}[Y|A = a, X]$, helping identify causal effects of interest which we detail below.

In **ATT estimation** [Ben-Michael et al., 2021], we are interested in the effect of a binary treatment on the population receiving it, that is $\mathbb{E}[Y(1) - Y(0)|A = 1]$. Thanks to consistency and no interference, $\mathbb{E}[Y(1)|A = 1]$ is accessible as the average of outcomes on the treated distribution, so the challenging part is estimating $\mathbb{E}[Y(0)|A = 1]$. The weighting approach is then to reweight the control distribution, on which $Y(0) = Y$, that is to find a function $w(x)$ such that

$$\mathbb{E}[Y(0)|A = 1] = \mathbb{E}[w(X)Y|A = 0] \approx \frac{1}{\{i : A_i = 0\}} \sum_{i:A_i=0} w(X_i)Y_i.$$

In **average potential outcome estimation** [Huling and Mak, 2024], for a fixed $a \in \mathcal{A}$, we are interested in the marginal effect of the potential outcome wrt $a$, that is $\mathbb{E}[Y(a)]$. The weighting approach is then to reweight the distribution of the population for which $A = a$, implying $Y(a) = Y$, i.e. find a function $w_a(x)$ such that

$$\mathbb{E}[Y(a)] = \mathbb{E}[w_a(X)Y|A = a] \approx \frac{1}{\{i : A_i = a\}} \sum_{i:A_i=a} w_a(X_i)Y_i.$$

We note that the closely related goal of **ATE estimation**, that is when $\mathcal{A}$ is binary and we want $\mathbb{E}[Y(1) - Y(0)]$, can be solved by average potential outcome estimation for both $a = 1$ and $a = 0$ separately. With some abuse of notation, we use the two names of average potential outcome estimation and ATE estimation interchangeably.

In **generalizability** and **transportability** [Colnet et al., 2024, Degtiar and Rose, 2023], $A$ is binary again and we have an other binary variable $S$ such that $S = 1$ denotes membership in the RCT population, that is $A \perp\!\!\!\perp X|S = 1$ and $(Y(1), Y(0)) \perp\!\!\!\perp A|S = 1$. We are interested in $\mathbb{E}[Y(1) - Y(0)]$ for generalizability and $\mathbb{E}[Y(1) - Y(0)|S = 0]$ for transportability. What motivates weighting here is that we do not have access to $A, Y$ when $S = 0$. Under the *transportability assumption*, the *conditional average treatment effect* is identical between RCT and non-RCT populations, i.e. for any $x$, $\text{CATE}(x) := \mathbb{E}[Y(1) - Y(0)|x]$ is equal to both $\mathbb{E}[Y(1) - Y(0)|x, S = 1]$ and $\mathbb{E}[Y(1) - Y(0)|x, S = 0]$. In addition, the CATE is identified on the RCT population as $\text{CATE}(x) = \mathbb{E}\left[\frac{AY}{P(A=1|S=1)} - \frac{(1-A)Y}{P(A=0|S=1)}\middle| X = x, S = 1\right]$. Then, defining $\pi = P(A = 1|S = 1)$, the weighting approach is to reweight the distribution of the RCT population, i.e. find weights $w$ such that

$$\mathbb{E}[Y(1) - Y(0)] = \mathbb{E}[w(X) \cdot \text{CATE}(X)|S = 1] \approx \frac{1}{|\{i : S_i = 1\}|} \sum_{i:S_i=1} w(X_i)\left(\frac{A_iY_i}{\pi} - \frac{(1-A_i)Y}{1-\pi}\right)$$

in generalizability or such that

$$\mathbb{E}[Y(1) - Y(0)|S = 0] = \mathbb{E}[w(X) \cdot \text{CATE}(X)|S = 1] \approx \frac{1}{|\{i : S_i = 1\}|} \sum_{i:S_i=1} w(X_i)\left(\frac{A_iY_i}{\pi} - \frac{(1-A_i)Y}{1-\pi}\right)$$

in transportability. Due to the similarity of both frameworks, without loss of generality, we focus on transportability as in Colnet et al. [2024] and Egami and Hartman [2021] which study variable selection in this setting.

Note that the framework of Problem 2.1 generally does not allow for **CATE estimation**, as the CATE is a function and the target $\mathbb{E}_Q[\mathbb{E}_P[\tilde{Y}|X]]$ is a scalar. Alternatively, one can perform simultaneous weightings as in Section 3.3, where for every problem we fix a covariate value $x_0$ and a treatment value $a$ and take the target estimand to be the $\mathbb{E}[Y|A = a, X = x_0]$. We can take the pseudo-outcome to be $Y$, the source distribution to be $P^{\text{data}}(X, Y|A = a)$ and the target distribution to be $P^{\text{data}}(X|X = x_0)$. However, this choice of target distribution would be a spike at $X = x_0$, potentially violating Assumption 2.3 in many widely-applicable situations, e.g. if the source distribution of covariates has a density wrt the Borel measure. As in Ben-Michael et al. [2021], such a problem with spikes could be mitigated with smoothing. If Assumption 2.3 does hold, e.g. if the source distribution of covariates is discrete and has a non-zero mass at $x_0$, then it could actually be possible to perform weighting, although we are not aware of such an approach in the previous literature.

# B    GENERALIZATION OF FORMER "SCORE" NOTIONS

More rigorously, the confounding bias is zero in three important cases:

1. $\mathbb{E}_P[\tilde{Y}|x]$ is a function of $\phi(x)$ $P_X-$a.s., where we call $\phi(x)$ a "generalized prognostic score" ;
2. $\frac{dQ}{dP}(x)$ is a function of $\phi(x)$ $P_X-$a.s., where we call $\phi(x)$ a "generalized balancing score" ;
3. The confounding bias is zero without $\phi$ necessarily being a generalized prognostic or balancing score, where we call $\phi(x)$ a "generalized deconfounding score".

The following result connects these notions to previous literature.

**Proposition B.1.** *In ATT/ATE estimation, a) balancing scores [Rosenbaum and Rubin, 1983b] are equivalent to generalized balancing scores . In ATE estimation, b) deconfounding scores [D'Amour and Franks, 2021] are equivalent to generalized deconfounding scores, c) prognostic scores [Hansen, 2008] are generalized prognostic scores, and the converse is true if*

$$\forall a \in \mathcal{A}, Y(a) \perp\!\!\!\perp X \mid \mathbb{E}[Y|X, A = a].$$

*In transportability [Egami and Hartman, 2021], assuming that the transportability assumption holds for X, d) heterogeneity sets are generalized prognostic scores, e) sampling sets are generalized balancing scores, f) separating sets are generalized deconfounding scores.*

Thus, these "generalized" scores extend existing notions of prognostic, balancing and deconfounding scores from the literature to the more general framework from Problem 2.1 and connect them to the confounding bias, refining our understanding of why these scores are well-suited for weighting. They also connect notions used for variable selection in transportability to the score notions from weighting for the ATT or the ATE.

We might say that the generalized notions clearly outline the "proper" definitions of their original counterparts, in a sense that they are either equivalent to them, or weaker than them while preserving properties required for deconfounding, as illustrated by generalized prognostic scores. Hence, for the remainder of the paper, we omit the "generalized" adjective from all notions of scores.

# C    REPRESENTATION SELECTION

To **select between two representations** $\phi_1$ and $\phi_2$, one can choose the representation with the lowest BSE. This is equivalent to compare $\min_{g_1} \mathcal{L}_{P,Q}(g_1(\phi_1(.)))$ and $\min_{g_2} \mathcal{L}_{P,Q}(g_2(\phi_2(.)))$, where each minimization is taken over all functions. These are inaccessible, but we can instead perform each minimization under a rich parameterized class of functions. Particularly, this would help select between two fitted propensity score models and we expect that the one with the best prediction performance might not necessarily be selected.

Further, we note that the AutoDML loss makes us lose the ability of evaluating how approximately deconfounding is *one* representation, instead of comparing different representations. Flexible density ratio estimators [Arbour et al., 2021] could be plugged into the balancing score error, especially as both the true weights and their expectation conditional on the representation are density ratios from Assumption 1 and Proposition 3.4.

# D   DETAILS ON EXPERIMENTS

**Code**   Our code is available at `https://github.com/oscarclivio/representations_weighting`.

**Origin of datasets**   We extracted IHDP [Hill, 2011] from the GitHub repository for Dragonnet [Shi et al., 2019] at `https://github.com/claudiashi57/dragonnet/tree/master/dat/ihdp/csv`, News [Johansson et al., 2016] from `https://www.fredjo.com/files/NEWS_csv.zip` and TBI [Colnet et al., 2024] from `https://github.com/BenedicteColnet/IPSW-categorical`. In addition, TBI is covered by a MIT license, and the original data source for News [Newman, 2008] by a CC BY 4.0 license.

**Infrastructure**   We ran experiments on a laptop with a GeForce GTX 1070 GPU with Max-Q Design and 12 CPU core. We used our own Python implementation for datasets (after downloading the data), weighting methods and representation learning techniques, including propensity score modelling and neural network fitting.

**Choice of hyperparameters**   We tried different sets of hyperparameters for neural networks, and first chose a set such that our approach had good performance (outperformed by at most one other method) on two different datasets each in a separate task under the energy balancing. Several sets verified this property, however performance of individual methods and individual hyperparameters was generally unequal among datasets. For ATE estimation, our method had the same ranking as in the paper for many hyperparameters on News, but was outperformed by standard kernel balancing on IHDP [Hill, 2011, Shalit et al., 2017] and ACIC 2016 [Dorie et al., 2019]. For ATT estimation, at the time of writing, we did not find such "good" hyperparameters on News and ACIC2016, but did so on IHDP and Jobs [LaLonde, 1986, Johansson et al., 2016]. This generally shows that different hyperparameters should be tested, especially for neural network-based methods. Defining a principled way (that does not use ground-truth target estimands) to select them for weighting has to be addressed in future work.

**Misspecified outcome classes**   We note that in our experiments, the outcome class $\mathcal{M}$ is often not correctly specified. Indeed, note that in our datasets

- The control outcome model in IHDP in its setting B [Hill, 2011], as used to generate the data [Shi et al., 2019], linearly depends on an exponential $e^{\beta \cdot x}$ function for some $\beta$.
- The outcome models for News linearly depend on the vector of topic probabilities $z(x)$ in Johansson et al. [2016], where we note that weights in this linear relationship are further positive. Noting $k$ the number of topics, we then have for any $x$, $z_i(x) \geq \frac{1}{k}$ for at least one $i$, and noting $w_0$ the minimal weight we obtain that $\forall a = 0, 1$, $\mathbb{E}[Y|x, A = a] \geq \frac{w_0}{k}$, thus either treated or control outcome model is bounded away from 0.
- The outcome model for TBI is quadratic in $x$ [Colnet et al., 2024].

Thus, outcome functions on IHDP, News and TBI are clearly not linear functions, thus misspecified for linear kernel optimal matching. For energy balancing, none of the outcome functions above is square-integrable (here *without* a probability measure; IHDP and TBI due to their functional forms, News due to being bounded away from $0$), thus none of them is Sobolev of any order. As the covariate space for IHDP and News has an odd dimensionality, these functions are misspecified outcome functions for the class corresponding to the energy distance according to page 12 of Mak and Joseph [2018]. This is less clear for the outcome model of the TBI dataset which has even dimensionality ; we conjecture that this outcome model is misspecified too, as its outcome model is not Sobolev of any order and Sobolev spaces up to a certain order are invoked as canonical members of the outcome class corresponding in Huling and Mak [2024].

# E   DETAILS ON OUR IMPLEMENTATION OF KERNEL OPTIMAL OR MEAN MATCHING

When applied to the representation $\phi$ composed through a kernel $k$, the square of the MMD is:

$$\text{MMD}_k^2(\mathcal{P}_w, \mathcal{Q}) = \frac{1}{|\mathcal{P}|^2} \sum_{i,j \in \mathcal{P}} w_i w_j k(\phi(x_i), \phi(x_j))$$
$$- \frac{2}{|\mathcal{P}||\mathcal{Q}|} \sum_{i \in \mathcal{P}, j \in \mathcal{P}} w_i k(\phi(x_i), \phi(x_j))$$

$$+ \frac{1}{|\mathcal{Q}|^2} \sum_{i,j \in \mathcal{Q}} k(\phi(x_i), \phi(x_j)).$$

Thus, its minimization with regularization is a quadratic problem (QP)

$$\min_w \frac{1}{2} w^T S w + v^T w \text{ subject to } l \leq A w \leq u \tag{9}$$

that can be solved with any off-the-shelf solver $\texttt{solver}(S, v, l, A, u)$ (e.g. Stellato et al. [2020]). Noting $I_\mathcal{P}$ the identity matrix over $\mathcal{P}$, we have

$$S = S_{\mathcal{P},\mathcal{Q}}^{k,\phi,\sigma} := (2/|\mathcal{P}|^2 \cdot k(\phi(x_i), \phi(x_j)) + 2\sigma^2 \cdot I_\mathcal{P})_{i,j \in \mathcal{P}},$$

$$v := v_{\mathcal{P},\mathcal{Q}}^{k,\phi} = (-2/|\mathcal{P}||\mathcal{Q}| \cdot \sum_{j \in \mathcal{Q}} k(\phi(x_i), \phi(x_j)))_{i \in \mathcal{P}},$$

$$A := A_\mathcal{P} = \left( \frac{I_\mathcal{P}}{1 \cdots 1} \right), \ l := l_\mathcal{P} = (\underbrace{0, \cdots, 0}_{|\mathcal{P}| \text{ times}}, |\mathcal{P}|)^T,$$

$$u = u_\mathcal{P} = (\underbrace{+\infty, \cdots, +\infty}_{|\mathcal{P}| \text{ times}}, |\mathcal{P}|)^T$$

For the joint squared bias with a finite $\Lambda = \{1, \cdots, \ell\}$, we sum all objectives from Proposition 3.3 over each $\alpha = 1, \cdots, \ell$ with $\mathcal{P}^\alpha, \mathcal{Q}^\alpha, \phi^\alpha, \sigma^\alpha$ and with kernel $k^\alpha$, giving

$$S = S_{\mathcal{P}^\Lambda, \mathcal{Q}^\Lambda}^{k^\Lambda, \phi^\Lambda, \sigma^\Lambda} := \text{diag}\left( (S_{\mathcal{P}^i, \mathcal{Q}^i}^{k^i, \phi^i, \sigma^i})_{i=1, \cdots, \ell} \right) \tag{10}$$

$$v = v_{\mathcal{P}^\Lambda, \mathcal{Q}^\Lambda}^{k^\Lambda, \phi^\Lambda} := \begin{pmatrix} v_{\mathcal{P}^1, \mathcal{Q}^1}^{k^1, \phi^1} \\ \vdots \\ v_{\mathcal{P}^\ell, \mathcal{Q}^\ell}^{k^\ell, \phi^\ell} \end{pmatrix}, \ \psi = \psi_{\mathcal{P}^\Lambda} := \begin{pmatrix} \psi_{\mathcal{P}^1} \\ \vdots \\ \psi_{\mathcal{P}^\ell} \end{pmatrix},$$

for $\psi \in A, l, u$. This step is agnostic to how $\sigma^\alpha$ is selected, either with a fixed value (e.g. 0 as in Huling and Mak [2024]) or from a principled procedure [Kallus, 2020b].pointwise

# F  PROOF OF RESULTS

## F.1  PROOF OF PROPOSITION 3.3

### F.1.1  Item 1

First, note that we can restrict our attention to weight functions $w$ in $L_2(P_X)$, that is such that $w(X) \in L_2(P)$, as the objective will be $\infty$ for weights functions not in $L_2(P_X)$. For any $L_2(P_X)$ weight function and function $g \in \mathcal{G}$, we have

$$\left| \mathbb{E}_{P_X^w}[g \circ \phi] - \mathbb{E}_{Q_X}[g \circ \phi] \right| = \left| \mathbb{E}_{P^w}[(g \circ \phi)(X)] - \mathbb{E}_Q[(g \circ \phi)(X)] \right|$$

$$= \left| \mathbb{E}_{P^w}[g(\phi(X))] - \mathbb{E}_Q[g(\phi(X))] \right|$$

$$= \left| \mathbb{E}_{P_{\phi(X)}^w}[g] - \mathbb{E}_{Q_{\phi(X)}}[g] \right|.$$

where all integrals are well-defined, as $g(\phi(X)) \in L_2(P)$ by assumption in the Proposition and $\frac{dQ_X}{dP_X}(X) \in L_2(P)$ from Assumptions 2.3 3.1. Taking the supremum over $g \in \mathcal{G}$, we have

$$\text{IPM}_\mathcal{M}(P_X^w, Q_X) = \text{IPM}_\mathcal{G}(P_{\phi(X)}^w, Q_{\phi(X)})$$

where $\mathcal{M} = \{x \to (g \circ \phi)(x), \ g \in \mathcal{G}\}$. Note that $\mathcal{M} \subseteq L_2(P_X)$, and that all of this also justifies the claim that the bias wrt $\phi$ is bounded by $\text{IPM}_\mathcal{G}(P_{\phi(X)}^w, Q_{\phi(X)})$. Thus, we are solving

$$\min_{w \in \mathcal{A}} J(w)$$

where

$$A := \{w \in L_2(P_X) \mid w \geq 0 \ P_X\text{-a.s.}, \mathbb{E}_P[w(X)] = 1\}$$
$$J(w) := I_{\mathcal{M}}(w)^2 + \sigma^2 \cdot S(w)$$
$$I_{\mathcal{M}}(w) := \mathrm{IPM}_{\mathcal{M}}(P_X^w, Q_X)$$
$$S(w) := \mathbb{E}_P[w(X)^2]$$

where functions in $L_2(P_X)$ are identified $P_X$-a.s.. We note that $\inf_{w \in \mathcal{A}} J(w)$ is finite, as $\frac{\mathrm{d}Q_X}{\mathrm{d}P_X} \in A$ from Assumption 3.1 and $J(\frac{\mathrm{d}Q_X}{\mathrm{d}P_X}) = \sigma^2 \cdot E_P\left[\left(\frac{\mathrm{d}Q_X}{\mathrm{d}P_X}(X)\right)^2\right] < \infty$.

We prove the first item of the Proposition in three parts :

1. There is at most one solution.

2. There is at least one solution.

3. Any solution is a function of $\phi(x)$ $P_X$-a.s.

Note that (i) only the third part uses the fact that functions in $\mathcal{M}$ are functions of $\phi(x)$ $P_X$-a.s., (ii) under stronger assumptions on the class $\mathcal{G}$, the result also follows directly from Theorem 4.1 of Bruns-Smith and Feller [2022], while the following analysis presents more relaxed assumptions over $\mathcal{G}$.

**Part 1 : There is at most one solution** $A$ is clearly a convex subset of $L_2(P_X)$, and $J$ is strictly convex. Indeed, for any $t \in [0,1]$, $w_1, w_2 \in A$, $m \in \mathcal{M}$, letting $w_t = tw_1 + (1-t)w_2$

$$|\mathbb{E}_{P^{w_t}}[m(X)] - \mathbb{E}_Q[m(X)]|$$
$$= |t\left(\mathbb{E}_{P^{w_1}}[m(X)] - \mathbb{E}_Q[m(X)]\right) + (1-t)\left(\mathbb{E}_{P^{w_2}}[m(X)] - \mathbb{E}_Q[m(X)]\right)|$$
$$\leq t|\mathbb{E}_{P^{w_1}}[m(X)] - \mathbb{E}_Q[m(X)]| + (1-t)|\mathbb{E}_{P^{w_2}}[m(X)] - \mathbb{E}_Q[m(X)]| \text{ from the triangle inequality}$$
$$\leq tI_{\mathcal{M}}(w_1) + (1-t)I_{\mathcal{M}}(w_2) \text{ taking the supremum wrt } m \text{ on each term on the RHS}$$

so taking the supremum wrt $m$ on the LHS, $I_{\mathcal{M}}(w_t) \leq tI_{\mathcal{M}}(w_1) + (1-t)I_{\mathcal{M}}(w_2)$ ; thus $I_{\mathcal{M}}$ is convex. As $u \mapsto u^2$ is convex non-decreasing, $I_{\mathcal{M}}^2$ is convex. Also, for any $t \in [0,1]$, $w_1, w_2 \in A$, $m \in \mathcal{M}$, again letting $w_t = tw_1 + (1-t)w_2$,

$$tS(w_1) + (1-t)S(w_2) - S(w_t)$$
$$= t\mathbb{E}_P[w_1(X)^2] + (1-t)\mathbb{E}_P[w_2(X)^2] - \mathbb{E}_P[(tw_1(X) + (1-t)w_2(X))^2]$$
$$= t\mathbb{E}_P[w_1(X)^2] + (1-t)\mathbb{E}_P[w_2(X)^2] - t^2\mathbb{E}_P[w_1(X)^2] - (1-t)^2\mathbb{E}_P[w_2(X)^2] - 2t(1-t)\mathbb{E}_P[w_1(X)w_2(X)]$$
$$= t(1-t)\mathbb{E}_P[(w_1(X) - w_2(X))^2]$$

which is non-negative, and zero iff $t = 0$, $t = 1$ or $w_1 = w_2$ $P_X-$a.s.. Thus, $S$ is strictly convex. Thus, the sum of $I_{\mathcal{M}}^2$ and $\sigma^2 \cdot S$, that is $J$, is strictly convex.

As $A$ is a convex subset of $L_2(P_X)$ and $J$ is strictly convex, there is at most one minimizer of $J$ in $A$.

**Part 2 : There is at least one solution.** From e.g. Theorem 2 of `https://www.math.umd.edu/~yanir/742/742-5-6.pdf`, the existence of a minimizer of $J$ in $A$ is guaranteed if $A$ is weakly closed and $J$ is coercive and sequentially weakly lower semi-continuous.

First, we show that $A$ is weakly closed. Let $w_n \in A^{\mathbb{N}}$ weakly converging to some $w_* \in L_2(P_X)$, that is such that

$$\forall h \in L_2(P_X), \ \mathbb{E}_P[w_n(X)h(X)] \xrightarrow[n \to \infty]{} \mathbb{E}_P[w_*(X)h(X)].$$

Then taking $h = 1$, we have $1 = \mathbb{E}_P[w_n(X)] \xrightarrow[n \to \infty]{} \mathbb{E}_P[w_*(X)]$, thus $\mathbb{E}_P[w_*(X)] = 1$.

Further, for $k \in \mathbb{N}^*$, let $B_k := \{w_*(X) \leq \frac{1}{k}\}$. Then, with $h := 1_{B_k}$, as $w_n \geq 0$ $P_X-$a.s. for each $n \in \mathbb{N}$

$$0 \leq \mathbb{E}_P[1_{B_k}(X)w_n(X)] \xrightarrow[n \to \infty]{} \mathbb{E}_P[w_*(X)h(X)] \leq -\frac{P(B_k)}{k}$$

which leads to $0 \leq \mathbb{E}_P[w_*(X)h(X)] \leq -\frac{P(B_k)}{k}$, which is not contradictory only if $P(B_k) = 0$. Then, as $\{w_*(X) < 0\} = \{\cup_{k \in \mathbb{N}^*} B_k\}$,

$$P(w_*(X) < 0) = P\left(\cup_{k \in \mathbb{N}^*} B_k\right)$$
$$\leq \sum_{k \in \mathbb{N}^*} P(B_k)$$
$$= 0$$

Thus, $w^* \geq 0$ $P_X$-a.s.. As a result, $w_* \in A$, so $A$ is weakly closed. We note that $J$ is coercive, as $S$ is clearly coercive and $I_{\mathcal{M}}$ is non-negative. What is left to prove in this part is then that $J$ is sequentially weakly lower semi-continuous. Let $w_n \in L_2(P_X)^{\mathbb{N}}$ weakly converging to some $w_* \in L_2(P_X)$. We want to show that

$$\liminf_{n \to \infty} J(w_n) \geq J(w_*).$$

Indeed,

$$\liminf_{n \to \infty} I_{\mathcal{M}}(w_n)^2 = \liminf_{n \to \infty} \sup_{m \in \mathcal{M}} |\mathbb{E}_P[w_n(X)m(X)] - \mathbb{E}_Q[m(X)]|$$
$$\geq \sup_{m \in \mathcal{M}} \liminf_{n \to \infty} \underbrace{|\mathbb{E}_P[w_n(X)m(X)] - \mathbb{E}_Q[m(X)]|^2}_{\substack{\xrightarrow[n \to \infty]{} |\mathbb{E}_P[w_*(X)m(X)] - \mathbb{E}_Q[m(X)]|^2 \\ \text{as } m \in L_2(P_X)}}$$
$$= \sup_{m \in \mathcal{M}} |\mathbb{E}_P[w_*(X)m(X)] - \mathbb{E}_Q[m(X)]|^2$$
$$= I_{\mathcal{M}}(w_*)^2$$

and by convexity of $u \mapsto u^2$,

$$\forall x, \ w_n(x)^2 \geq w_*(x)^2 + 2w_*(x)(w_n(x) - w_*(x))$$

so

$$\liminf_{n \to \infty} S(w_n) = \liminf_{n \to \infty} \mathbb{E}_P[w_n(X)^2]$$
$$\geq \liminf_{n \to \infty} \mathbb{E}_P[w_*(X)^2] + 2\big(\underbrace{\mathbb{E}_P[w_n(X)w_*(X)] - \mathbb{E}_P[w_*(X)^2]}_{\xrightarrow[n \to \infty]{} 0}\big)$$
$$= \mathbb{E}_P[w_*(X)^2]$$
$$= S(w_*)$$

and

$$\liminf_{n \to \infty} I_{\mathcal{M}}(w_n)^2 + \sigma^2 S(w_n) \geq \liminf_{n \to \infty} I_{\mathcal{M}}(w_n) + \liminf_{n \to \infty} \sigma^2 S(w_n)$$
$$\geq I_{\mathcal{M}}(w_*)^2 + \sigma^2 S(w_*) \text{ from the above.}$$

All of this shows that $J$ is sequentially weakly lower semi-continuous, concluding this part of the proof.

**Part 3 : Any solution is a function of** $\phi(x)$. For any $w \in L_2(P_X)$, let $\bar{w}(z) = \mathbb{E}_P[w(X)|\phi(X) = z]$. If $w \in A$, then $\bar{w}(\phi(.)) \in A$. Indeed, the conditional expectation of any $L_2(P)$ random variable is also $L_2(P)$, so $\bar{w}(\phi(.)) \in L_2(P_X)$. Further, the conditional expectation of any almost surely non-negative random variable is also almost surely non-negative, so $\bar{w}(\phi(.)) \geq 0$ $P_X$-a.s.. Finally, the tower property shows that

$$\mathbb{E}_P[\bar{w}(\phi(X))] = \mathbb{E}_P[\mathbb{E}_P[w(X)|\phi(X)]] = \mathbb{E}_P[w(X)] = 1.$$

Thus, $\bar{w}(\phi(.)) \in A$. It actually turns out that $J(\bar{w}(\phi(.))) \leq J(w)$, with equality iff $w = \bar{w}(\phi(.))$. This concludes the proof, as a minimizer of $J$ in $A$ has to be a function of $\phi(x)$, as otherwise we can construct a weight function in $A$ that realises a strictly lower objective, which is contradictory.

First,

$$\forall g \in \mathcal{G}, \ \mathbb{E}_P[w(X)g(\phi(X))] = \mathbb{E}_P[\mathbb{E}_P[w(X)g(\phi(X))|\phi(X)]] \text{ from the tower property}$$
$$= \mathbb{E}_P[\mathbb{E}_P[w(X)|\phi(X)]g(\phi(X))]$$
$$= \mathbb{E}_P[\bar{w}(\phi(X))g(\phi(X))]$$

so $I_{\mathcal{M}}(\bar{w}(\phi(.))) = I_{\mathcal{M}}(w)$. Further,

$$S(w) - S(\bar{w}(\phi(.))) = \mathbb{E}_P[w(X)^2] - \mathbb{E}_P[\mathbb{E}_P[w(X)|\phi(X)]^2]$$
$$= \mathbb{E}_P[\mathbb{E}_P[w(X)^2|\phi(X)]] - \mathbb{E}_P[\mathbb{E}_P[w(X)|\phi(X)]^2] \text{ from the tower property}$$
$$= \mathbb{E}_P\left[\mathbb{E}_P[w(X)^2|\phi(X)]] - \mathbb{E}_P[w(X)|\phi(X)]^2\right]$$
$$= \mathbb{E}_P[\mathrm{Var}(w(X)|\phi(X))]$$
$$= \mathbb{E}_P[\mathbb{E}_P[(w(X) - \bar{w}(\phi(X)))^2|\phi(X)]]$$
$$= \mathbb{E}_P[(w(X) - \bar{w}(\phi(X)))^2] \text{ from the tower property.}$$

Taken all together, $J(w) \geq J(\bar{w}(\phi(.)))$ with equality iff $\mathbb{E}_P[(w(X) - \bar{w}(\phi(X)))^2] = 0$, that is $w = \bar{w}(\phi(.)) \ P_X$-a.s.. This concludes the proof.

### F.1.2   Item 2

Let $w$ be an $L_2(P_X)$ weight function such that $w = \bar{w}(\phi(.)) \ P_X$-a.s. for some $\bar{w}$. Then,

$$\text{Chosen weights bias of } w = \mathbb{E}_{P^w}\left[\mathbb{E}_P[\tilde{Y}|X] - \mathbb{E}_P[\tilde{Y}|\phi(X)]\right]$$
$$= \mathbb{E}_P\left[w(X)\mathbb{E}_P[\tilde{Y}|X] - w(X)\mathbb{E}_P[\tilde{Y}|\phi(X)]\right]$$
$$= \mathbb{E}_P\left[\bar{w}(\phi(X))\mathbb{E}_P[\tilde{Y}|X] - \bar{w}(\phi(X))\mathbb{E}_P[\tilde{Y}|\phi(X)]\right]$$
$$= \mathbb{E}_P\left[\bar{w}(\phi(X))\mathbb{E}_P[\tilde{Y}|X]\right] - \mathbb{E}_P\left[\bar{w}(\phi(X))\mathbb{E}_P[\tilde{Y}|\phi(X)]\right]$$
$$= \mathbb{E}_P\left[\bar{w}(\phi(X))\mathbb{E}_P[\tilde{Y}|X]\right] - \mathbb{E}_P\left[\bar{w}(\phi(X))\mathbb{E}_P[\tilde{Y}|\phi(X)]\right]$$
$$= \mathbb{E}_P\left[\mathbb{E}_P[\bar{w}(\phi(X))\tilde{Y}|X]\right] - \mathbb{E}_P\left[E_P[\bar{w}(\phi(X))\tilde{Y}|\phi(X)]\right]$$
$$= \mathbb{E}_P[\bar{w}(\phi(X))\tilde{Y}] - E_P[\bar{w}(\phi(X))\tilde{Y}] \text{ from the tower property}$$
$$= 0$$

### F.2   PROOF OF PROPOSITION 3.4

Let $\Sigma_Z$ denote the $\sigma$-algebra of the space of values taken by random variable $Z$.

Let $B \in \Sigma_{\phi(X)}$ such that $P_{\phi(X)}(B) = 0$. Then $0 = P_{\phi(X)}(B) = P_X(\phi^{-1}(B))$ where $\phi^{-1}(B) \in \Sigma_X$ as $\phi$ is measurable. By Assumption 2.3, $Q_X(\phi^{-1}(B)) = 0$. Then $0 = Q_X(\phi^{-1}(B)) = Q_{\phi(X)}(B)$. Thus, $Q_{\phi(X)}$ is absolutely continuous wrt $P_{\phi(X)}$.

Notably, from the Radon-Nikodym theorem, $\frac{\mathrm{d}Q_{\phi(X)}}{\mathrm{d}P_{\phi(X)}}$ exists. Then for any $B \in \Sigma_{\phi(X)}$,

$$\mathbb{E}_P\left[\frac{\mathrm{d}Q_{\phi(X)}}{\mathrm{d}P_{\phi(X)}}(\phi(X)) \cdot 1_B(\phi(X))\right]$$
$$= \mathbb{E}_Q[1_B(\phi(X))]$$
$$= \mathbb{E}_P\left[\frac{\mathrm{d}Q_X}{\mathrm{d}P_X}(X) \cdot 1_B(\phi(X))\right] \text{ by taking the Radon-Nikodym derivative wrt } X$$
$$= \mathbb{E}_P\left[\mathbb{E}_P\left[\frac{\mathrm{d}Q_X}{\mathrm{d}P_X}(X) \cdot 1_B(\phi(X))\Big|\phi(X)\right]\right] \text{ from the tower property}$$

$$= \mathbb{E}_P\left[\mathbb{E}_P\left[\frac{\mathrm{d}Q_X}{\mathrm{d}P_X}(X)\Big|\phi(X)\right]\cdot 1_B(\phi(X))\right]$$

where all integrals are well-defined as the Radon-Nikodym derivative is measurable and $L_1(P_X)$, and its conditional expectation is also $L_1(P_{\phi(X)})$ as any conditional expectation of any $L_1(P)$ random variable is also $L_1(P)$.

Thus we have shown that $\forall B \in \Sigma_{\phi(X)}$, $\int h \cdot 1_B \mathrm{d}P_{\phi(X)} = 0$ where $h(z) = \frac{\mathrm{d}Q_{\phi(X)}}{\mathrm{d}P_{\phi(X)}}(z) - \mathbb{E}_P\left[\frac{\mathrm{d}Q_X}{\mathrm{d}P_X}(X)\Big|\phi(X) = z\right]$. We now show that $h = 0$, which concludes the proof for the first part of the Proposition. Note that $h$ is measurable as any Radon-Nikodym derivative is measurable, and any conditional expectation is measurable. Notably, as $\mathbb{R}_+$ and $\mathbb{R}_-$ are in the Borel $\sigma$-algebra, $B_+ = h^{-1}(\mathbb{R}_+)$ and $B_- = h^{-1}(\mathbb{R}_-)$ are in $\Sigma_{\phi(X)}$. Thus,

$$0 = \int_{\mathcal{Z}} h \cdot 1_{B_+}\mathrm{d}P_{\phi(X)} = \int_{\mathcal{Z}} h_+\mathrm{d}P_{\phi(X)}$$

$$0 = \int_{\mathcal{Z}} h \cdot 1_{B_-}\mathrm{d}P_{\phi(X)} = -\int_{\mathcal{Z}} h_-\mathrm{d}P_{\phi(X)}$$

which implies that $h_+ = 0$ and $h_- = 0$, both $P_{\phi(X)}$-a.s., as these two functions are non-negative. Thus, $h = 0$ $P_{\phi(X)}$-a.s., which concludes the first part of proof.

Now we further assume Assumptions 3.1 and 3.2. Then, we note that the confounding bias is equal to $-\mathbb{E}_P\left[\frac{\mathrm{d}Q_X}{\mathrm{d}P_X}(X)\left(\mathbb{E}_P[\tilde{Y}|X] - \mathbb{E}_P[\tilde{Y}|\phi(X)]\right)\right]$. As $\frac{\mathrm{d}Q_X}{\mathrm{d}P_X}$ is now a $L_2(P_X)$ weight function wrt $P$, and using that $\frac{\mathrm{d}Q_{\phi(X)}}{\mathrm{d}P_{\phi(X)}} = \mathbb{E}_P\left[\frac{\mathrm{d}Q_X}{\mathrm{d}P_X}(X)\Big|\phi(X) = .\right]$ $P_{\phi(X)}$-a.s., identical computations as in the proof of item 2 in Proposition 3.3 show that $\frac{\mathrm{d}Q_{\phi(X)}}{\mathrm{d}P_{\phi(X)}}(\phi(.))$ is also a $L_2(P_X)$ weight function wrt $P$, while being a function of $\phi(x)$. Applying Proposition 3.3, item 2, to $\frac{\mathrm{d}Q_{\phi(X)}}{\mathrm{d}P_{\phi(X)}}(\phi(.))$ leads to $\mathbb{E}_P\left[\frac{\mathrm{d}Q_{\phi(X)}}{\mathrm{d}P_{\phi(X)}}(\phi(X))\left(\mathbb{E}_P[\tilde{Y}|X] - \mathbb{E}_P[\tilde{Y}|\phi(X)]\right)\right] = 0$. Summing this to the confounding bias leads to

$$\text{Confounding bias} = -\mathbb{E}_P\left[\left(\frac{\mathrm{d}Q_X}{\mathrm{d}P_X}(X) - \frac{\mathrm{d}Q_{\phi(X)}}{\mathrm{d}P_{\phi(X)}}(\phi(X))\right)\cdot\left(\mathbb{E}_P[\tilde{Y}|X] - \mathbb{E}_P[\tilde{Y}|\phi(X)]\right)\right].$$

Finally,

$$\mathbb{E}_P\left[\left(\frac{\mathrm{d}Q_X}{\mathrm{d}P_X}(X) - \frac{\mathrm{d}Q_{\phi(X)}}{\mathrm{d}P_{\phi(X)}}(\phi(X))\right)\mathbb{E}_P[\tilde{Y}|\phi(X)]\right]$$

$$= \mathbb{E}_P\left[\left(\frac{\mathrm{d}Q_X}{\mathrm{d}P_X}(X) - \mathbb{E}_P\left[\frac{\mathrm{d}Q_X}{\mathrm{d}P_X}(X)\Big|\phi(X)\right]\right)\mathbb{E}_P[\tilde{Y}|\phi(X)]\right] \text{ from the first part of the Proposition}$$

$$= \mathbb{E}_P\left[\frac{\mathrm{d}Q_X}{\mathrm{d}P_X}(X)\mathbb{E}_P[\tilde{Y}|\phi(X)]\right] - \mathbb{E}_P\left[\mathbb{E}_P\left[\frac{\mathrm{d}Q_X}{\mathrm{d}P_X}(X)\Big|\phi(X)\right]\mathbb{E}_P[\tilde{Y}|\phi(X)]\right]$$

$$= \mathbb{E}_P\left[\frac{\mathrm{d}Q_X}{\mathrm{d}P_X}(X)\mathbb{E}_P[\tilde{Y}|\phi(X)]\right] - \mathbb{E}_P\left[\mathbb{E}_P\left[\frac{\mathrm{d}Q_X}{\mathrm{d}P_X}(X)\mathbb{E}_P[\tilde{Y}|\phi(X)]\Big|\phi(X)\right]\right]$$

$$= \mathbb{E}_P\left[\frac{\mathrm{d}Q_X}{\mathrm{d}P_X}(X)\mathbb{E}_P[\tilde{Y}|\phi(X)]\right] - \mathbb{E}_P\left[\frac{\mathrm{d}Q_X}{\mathrm{d}P_X}(X)\mathbb{E}_P[\tilde{Y}|\phi(X)]\right] \text{ from the tower property}$$

$$= 0$$

Thus,

$$\text{Confounding bias} = -\mathbb{E}_P\left[\left(\frac{\mathrm{d}Q_X}{\mathrm{d}P_X}(X) - \frac{\mathrm{d}Q_{\phi(X)}}{\mathrm{d}P_{\phi(X)}}(\phi(X))\right)\cdot\mathbb{E}_P[\tilde{Y}|X]\right].$$

## F.3  PROOF OF COROLLARY 3.5

Note that from the tower property,

$$\mathbb{E}_P[\tilde{Y}|\phi(X)] = \mathbb{E}_P[\mathbb{E}_P[\tilde{Y}|X,\phi(X)]|\phi(X)] = \mathbb{E}_P[\mathbb{E}_P[\tilde{Y}|X]\mid\phi(X)] \tag{11}$$

From Proposition 3.3, for any $w$ depending on $\phi$ $P_X$-a.s., the zero chosen weights bias is zero. Thus,

$$|\text{Bias}_{P,Q}(w)| \leq \left|\mathbb{E}_{P^w}[\mathbb{E}[\tilde{Y}|\phi(X)]] - \mathbb{E}_Q[\mathbb{E}[\tilde{Y}|\phi(X)]]\right| + |\text{Confounding bias}|$$

$$\text{where } \mathbb{E}_P[\tilde{Y}|x] \in \mathcal{M} \text{ so from Equation 11, } \mathbb{E}_P[\tilde{Y}|\phi(x)] \in \phi(\mathcal{M}, P)$$

$$\leq \text{IPM}_{\phi(\mathcal{M},P)}(P_{\phi(X)}^w, Q_{\phi(X)}) + |\text{Confounding bias}| \text{ by definition of an IPM}$$

$$\leq \text{IPM}_{\phi(\mathcal{M},P)}(P_{\phi(X)}^w, Q_{\phi(X)}) + ||\tilde{Y}||_{L_2(P)} \cdot \text{BSE}_{P,Q}(\phi) \text{ from Equation 7}$$

## F.4 PROOF OF COROLLARY 3.6

From Corollary 3.5, for any $\alpha \in \Lambda$,

$$\text{Bias}_{P^\alpha,Q^\alpha}^2(w^\alpha)$$
$$\leq \left(\text{IPM}_{\phi^\alpha(\mathcal{M}^\alpha,P^\alpha)}(P_{\phi^\alpha(X)}^{\alpha,w^\alpha}, Q_{\phi^\alpha(X)}^\alpha) + ||\tilde{Y}||_{L_2(P^\alpha)} \cdot \text{BSE}_{P^\alpha,Q^\alpha}(\phi^\alpha)\right)^2.$$

Noting that $\forall a, b, (a+b)^2 \leq 2(a^2 + b^2)$ and taking the expectation wrt $p_\Lambda(\alpha)$ gives

$$\frac{1}{2} \cdot \text{Bias}_{P^\Lambda,Q^\Lambda}^2(w^\Lambda) \leq \mathbb{E}_{p_\Lambda(\alpha)}\left[\text{IPM}_{\phi^\alpha(\mathcal{M}^\alpha,P^\alpha)}^2(P_{\phi^\alpha(X)}^{\alpha,w^\alpha}, Q_{\phi^\alpha(X)}^\alpha)\right] + \mathbb{E}_{p_\Lambda(\alpha)}\left[||\tilde{Y}||_{L_2(P^\alpha)}^2 \cdot \text{BSE}_{P^\alpha,Q^\alpha}^2(\phi^\alpha)\right]$$

Taking $||\tilde{Y}||_{L_2(P^\alpha)}^2 \leq \sup_{\alpha \in \Lambda} ||\tilde{Y}||_{L_2(P^\alpha)}^2$ in the expectation with the BSE's leads to the result.

## F.5 PROOF OF PROPOSITION B.1

First, let's note two useful properties :

- For any distribution $R$ and random variable $Z$,

$$\forall x, \quad \mathbb{E}_R[\mathbb{E}_R[Z|X] \mid \phi(X) = \phi(x)] = \mathbb{E}_R[Z|\phi(X) = \phi(x)]. \tag{12}$$

- For any distributions $R$ and function $f$,

$$\left(\exists g, \ \forall x \ R_X\text{-a.s., } f(x) = g(\phi(x))\right) \Leftrightarrow \forall x \ R_X\text{-a.s., } f(x) = \mathbb{E}_R[f(X) \mid \phi(X) = \phi(x)]. \tag{13}$$

**Proof of a), ATT case :** Let $e(x) := P^{\text{data}}(A = 1|X = x)$

$\phi$ is a balancing score

$\Leftrightarrow \exists g, \ e(x) = g(\phi(x)) \ \forall x \ P_X^{\text{data}}$-a.s. from [Rosenbaum and Rubin, 1983b]

$\Leftrightarrow \exists g, \ e(x) = g(\phi(x)) \ \forall x \ P_{X|A=0}^{\text{data}}$-a.s. from the overlap assumption

$\Leftrightarrow \exists g, \ \dfrac{\text{d}P_{X|A=1}^{\text{data}}}{\text{d}P_{X|A=0}^{\text{data}}}(x) = g(\phi(x)) \ \forall x \ P_{X|A=0}^{\text{data}}$-a.s. as $\dfrac{\text{d}P_{X|A=1}^{\text{data}}}{\text{d}P_{X|A=0}^{\text{data}}}(x)$ is a bijective function of $e(x)$ from Bayes' rule

$\Leftrightarrow \phi(x)$ is a generalized balancing score.

**Proof of a), ATE case** : we fix $a \in \mathcal{A}$ and work with the following definition [Imbens, 2000] of a balancing score for non-binary treatments : $1_{\{A=a\}} \perp\!\!\!\perp X|\phi(X)$. Indeed, as the problem is arm-specific, the definitions of generalized deconfounding, balancing and prognostic scores are arm-specific *a priori*. An extension to an alternative definition $A \perp\!\!\!\perp X|\phi(X)$ is straightforward by replacing a fixed $a \in \mathcal{A}$ with $\forall a \in \mathcal{A}$ at the start of each of the following statements involving $a$. Then,

$\phi$ is a balancing score

---

[1]While the original statement in Rosenbaum and Rubin [1983b] is not $P_X^{\text{data}}$-a.s., we note that it can be relaxed to $P_X^{\text{data}}$-a.s. as it pertains to the adjustment formula that involves an expectation wrt $P_X^{\text{data}}$

$$\Leftrightarrow P^{\text{data}}(a|x) = P^{\text{data}}(a|\phi(x)) \; \forall x \; P_X^{\text{data}}\text{-a.s.}$$

$$\Leftrightarrow P^{\text{data}}(a|x) = \mathbb{E}[P^{\text{data}}(a|X)|\phi(X) = \phi(x)] \; \forall x \; P_X^{\text{data}}\text{-a.s. using 12 with } Z = 1_{\{A=a\}}$$

$$\Leftrightarrow \exists g_a, \;\; P^{\text{data}}(a|x) = g_a(\phi(x)) \; \forall x \; P_X^{\text{data}}\text{-a.s. from 13}$$

$$\Leftrightarrow \exists g_a, \;\; \frac{dP_X^{\text{data}}}{dP_{X|A=a}^{\text{data}}}(x) = g_a(\phi(x)) \; \forall x \; P_X^{\text{data}}\text{-a.s.}$$

where $\dfrac{dP_X^{\text{data}}}{dP_{X|A=a}^{\text{data}}}(x)$ is the true weights and is a bijective function of $P^{\text{data}}(a|x)$ from Bayes' rule

$$\Leftrightarrow \exists g_a, \;\; \frac{dP_X^{\text{data}}}{dP_{X|A=a}^{\text{data}}}(x) = g_a(\phi(x)) \; \forall x \; P_{X|A=a}^{\text{data}}\text{-a.s. from the overlap assumption}$$

$$\Leftrightarrow \phi(x) \text{ is a generalized balancing score.}$$

**Proof of b)** : we slightly change the definition of deconfounding scores [D'Amour and Franks, 2021] to $\forall a \in \mathcal{A}, \; \mathbb{E}[\mathbb{E}[Y|\phi(X), A = a]] = \mathbb{E}[Y(a)]$, where the representation $\phi$ is now shared across treatment arms, in the spirit of D'Amour and Franks (2021)[D'Amour and Franks, 2021].To this aim, it is sufficient to show that, in Problem 2.1 applied to estimation of $\mathbb{E}[Y(a)]$, the confounding bias is equal to $\mathbb{E}[\mathbb{E}[Y|\phi(X), A = a]] - \mathbb{E}[Y(a)]$. From the original definition of the confounding bias, this simplifies further to $\mathbb{E}[Y(a)] = \mathbb{E}[\mathbb{E}[Y|X, A = a]]$. This follows from the canonical unconfoundedness, overlap and SUTVA assumptions.

**Proof of c)** : again, $a \in \mathcal{A}$ is fixed. Assume $\phi(x)$ is a prognostic score for $Y(a)$, that is $Y(a) \perp\!\!\!\perp X|\phi(X)$. Then,

$$\forall x \; P_{X|A=a}^{\text{data}}\text{-a.s.}, \mathbb{E}[Y|x, A = a] := \mathbb{E}[Y(a)|x]$$
$$= \mathbb{E}[Y(a)|x, \phi(x)]$$
$$= \mathbb{E}[Y(a)|\phi(x)] \text{ by application of the definition of a prognostic score,}$$

so $\mathbb{E}[Y|x, A = a]$ is a function of $\phi(x) \; P_X^{\text{data}}$-a.s., thus it is so $P_{X|A=a}^{\text{data}}$-a.s. from the overlap assumption, making the latter a generalized prognostic score.

Now assume that $\mathbb{E}[Y|x, A = a]$ itself is a prognostic score, that is $Y(a) \perp\!\!\!\perp X \mid \mathbb{E}[Y|X, A = a]$. Then, $p^{\text{data}}(Y(a)|x) = p^{\text{data}}(Y(a)|\mathbb{E}[Y|x, A = a]) \; \forall x \; P_X^{\text{data}}$-a.s., where $P_X^{\text{data}}$-a.s. can be replaced with $P_{X|A=a}^{\text{data}}$-a.s. thanks to the overlap assumption. Let $\phi(X)$ be a generalized prognostic score. Then, there exists a function $g_a$ such that $\mathbb{E}[Y|x, A = a] = g_a(\phi(x)) \; \forall x \; P_{X|A=a}^{\text{data}}$-a.s.. In particular, as $p^{\text{data}}(Y(a)|x)$ is already a function of $\mathbb{E}[Y|x, A = a] \; P_{X|A=a}^{\text{data}}$-a.s., it is also a function of $\phi(x) \; P_{X|A=a}^{\text{data}}$-a.s.. So there exists a function $h_a$ such that $p^{\text{data}}(Y(a)|x) = h_a(\phi(x)) \; \forall x \; P_{X|A=a}^{\text{data}}$-a.s.. In particular, by application of 13, $p^{\text{data}}(Y(a)|x) = \mathbb{E}[p^{\text{data}}(Y(a)|X)|\phi(X) = \phi(x)] \; \forall x \; P_{X|A=a}^{\text{data}}$-a.s. and by application of 12 to $Z = 1_{\{Y(a)=.\}}$, $p^{\text{data}}(Y(a)|x) = p^{\text{data}}(Y(a)|\phi(x)) \; \forall x \; P_{X|A=a}^{\text{data}}$-a.s., which can be replaced with $P_X^{\text{data}}$-a.s. from the overlap assumption. Thus, $\phi(x)$ is a prognostic score.

**Proof of d)** : let $X_{\mathcal{I}}$ be covariates selected according to indices $\mathcal{I}$ and $X_{-\mathcal{I}}$ be their complement. We also use this notation for e) and f).

If $x_{\mathcal{I}}$ is a heterogeneity set, i.e. $Y(1) - Y(0) \perp\!\!\!\perp (S, X_{-\mathcal{I}})|X_{\mathcal{I}}$ then

$$\forall x \; P_X^{\text{data}}\text{-a.s.}, \mathbb{E}_P[\tilde{Y}|x] = \text{CATE}(x) \text{ (under the transportability assumption)}$$
$$= \mathbb{E}[Y(1) - Y(0)|x]$$
$$= \mathbb{E}[Y(1) - Y(0)|x_{-\mathcal{I}}, x_{\mathcal{I}}]$$
$$= \mathbb{E}[Y(1) - Y(0)|x_{\mathcal{I}}] \text{ by definition of a heterogeneity set}$$

where $P_X^{\text{data}}$-a.s. is equivalent to $P_{X|S=1}^{\text{data}}$-a.s. under the support inclusion (i.e. overlap) assumption, so $\mathbb{E}_P[\tilde{Y}|x]$ is a function of $x_{\mathcal{I}} \; P_{X|S=1}^{\text{data}}$-a.s., making the latter a generalized prognostic score.

**Proof of e)** : If $x_{\mathcal{I}}$ is a sampling set, that is $Y(1), Y(0), S \perp\!\!\!\perp X_{-\mathcal{I}}|X_{\mathcal{I}}$, then

$$\forall x P_X^{\text{data}}\text{-a.s.}, \frac{dP_{X|S=0}^{\text{data}}}{dP_{X|S=1}^{\text{data}}}(x) = \frac{p^{\text{data}}(x|S = 0)}{p^{\text{data}}(x|S = 1)}$$

$$
\begin{aligned}
&= \frac{P^{\text{data}}(S=1)}{P^{\text{data}}(S=0)} \frac{p^{\text{data}}(S=0|x)}{p^{\text{data}}(S=1|x)} \text{ from Bayes' rule} \\
&= \frac{P^{\text{data}}(S=1)}{P^{\text{data}}(S=0)} \frac{p^{\text{data}}(S=0|x_{\mathcal{I}}, x_{-\mathcal{I}})}{p^{\text{data}}(S=1|x_{\mathcal{I}}, x_{-\mathcal{I}})} \\
&= \frac{P^{\text{data}}(S=1)}{P^{\text{data}}(S=0)} \frac{p^{\text{data}}(S=0|x_{\mathcal{I}})}{p^{\text{data}}(S=1|x_{\mathcal{I}})} \text{ as } x_{\mathcal{I}} \text{ is a sampling set} \\
&= \frac{p^{\text{data}}(x_{\mathcal{I}}|S=0)}{p^{\text{data}}(x_{\mathcal{I}}|S=1)} \\
&= \frac{\mathrm{d}P^{\text{data}}_{X_{\mathcal{I}}|S=0}}{\mathrm{d}P^{\text{data}}_{X_{\mathcal{I}}|S=1}}(x_{\mathcal{I}})
\end{aligned}
$$

thus $\frac{\mathrm{d}P^{\text{data}}_{X|S=0}}{\mathrm{d}P^{\text{data}}_{X|S=1}}(x)$ depends on $x_{\mathcal{I}} \ \forall x \ P^{\text{data}}_X$-a.s., which is equivalent to $P^{\text{data}}_{X|S=1}$-a.s. under the support inclusion (i.e. overlap) assumption, and the last two lines illustrate the fact that, in this case, the density ratio wrt $X$ is equal to that wrt the representation a.s. under the source distribution.

**Proof of f)** : If $x_{\mathcal{I}}$ is a separating set, that is $Y(1) - Y(0) \perp\!\!\!\perp S|X_{\mathcal{I}}$,

$$
\begin{aligned}
P^{\text{data}}\text{-a.s., } \mathbb{E}_P[\tilde{Y}|X_{\mathcal{I}}, S=0] &= \mathbb{E}_P[\mathbb{E}_P[\tilde{Y}|X]|X_{\mathcal{I}}] \\
&= \mathbb{E}_P[\text{CATE}(X)|X_{\mathcal{I}}] \\
&= \mathbb{E}[\text{CATE}(X)|X_{\mathcal{I}}, S=1] \\
&= \mathbb{E}[\mathbb{E}[Y(1)-Y(0)|X, S=1]|X_{\mathcal{I}}, S=1] \text{ under the transportability assumption} \\
&= \mathbb{E}[\mathbb{E}[Y(1)-Y(0)|X, X_{\mathcal{I}}, S=1]|X_{\mathcal{I}}, S=1] \\
&= \mathbb{E}[Y(1)-Y(0)|X_{\mathcal{I}}, S=1] \text{ under the tower property} \\
&= \mathbb{E}[Y(1)-Y(0)|X_{\mathcal{I}}, S=0] \text{ by definition of a separating set.}
\end{aligned}
$$

where $P^{\text{data}}$-a.s. implies $P^{\text{data}}(.|S=0) - $ a.s., thus

Confounding bias of $x_{\mathcal{I}}$

$$
\begin{aligned}
&= \mathbb{E}_Q\left[\mathbb{E}_P[\tilde{Y}|X_{\mathcal{I}}] - \mathbb{E}_P[\tilde{Y}|X]\right] \\
&= \mathbb{E}\left[\mathbb{E}[Y(1)-Y(0)|X_{\mathcal{I}}, S=0] - \mathbb{E}_P[\tilde{Y}|X]\Big|S=0\right] \text{ from the above} \\
&= \mathbb{E}\left[\mathbb{E}[Y(1)-Y(0)|X_{\mathcal{I}}, S=0] - \text{CATE}(X)\Big|S=0\right] \\
&= \mathbb{E}\left[\mathbb{E}[Y(1)-Y(0)|X_{\mathcal{I}}, S=0] - \mathbb{E}[Y(1)-Y(0)|X, S=0]\Big|S=0\right] \text{ from the transportability assumption} \\
&= \mathbb{E}\left[\mathbb{E}[Y(1)-Y(0)|X_{\mathcal{I}}, S=0]\Big|S=0\right] - \mathbb{E}\left[\mathbb{E}[Y(1)-Y(0)|X, S=0]\Big|S=0\right] \\
&= \mathbb{E}[Y(1)-Y(0)|S=0] - \mathbb{E}[Y(1)-Y(0)|S=0] \text{ under the tower property} \\
&= 0,
\end{aligned}
$$

so $x_{\mathcal{I}}$ is a generalized deconfounding score.