# OpenReview forum: "Towards Representation Learning for Weighting Problems in Design-Based Causal Inference"
_auai.org/UAI/2024/Conference — UAI 2024 poster_

### Official Review · Reviewer_wiXK · 2024-03-01

**Q2-1 Originality-Novelty:** 2
**Q2-2 Correctness-Technical Quality:** 2
**Q2-5 Clarity Of Writing:** 3

**Q1 Summary And Contributions:**

The authors develop a weighting approach for treatment effect estimation. They theoretically analyze the bias term and explicitly formulate the « confounding bias » term, which is used to learn a better representation for treatment effect estimation. The proposed method seems theoretically sound. The paper would be stronger if there are further experimental results.

**Q2-3 Extent To Which Claims Are Supported By Evidence:**

3: Good: the main claims are supported by convincing evidence (in the form of adequate experimental evaluation, proofs, (pseudo-)code, references, assumptions).

**Q2-4 Reproducibility:**

3: Good: key resources (e.g. proofs, code, data) are available and key details (e.g. proofs, experimental setup) are sufficiently well-described for competent researchers to confidently reproduce the main results.

**Q3 Main Strengths:**

- The proposed method seems technically sound.

- The weighting approach is very common in causal inference, so the theoretical results seem solid and strong.

**Q4 Main Weakness:**

(A) Experimental results

- Although the proposed weighting approach can deal with various treatment-effect statistics, the authors only evaluate the performance of average treatment effect (ATE) estimation in Section 5. I believe that the proposed method can address CATE estimation (according to Appendix ). If so, it would be better if the authors perform additional experiments.

(B) The motivation

- In Section 1, the authors mention the importance of addressing « outcome-agnostic » weights, which are not formulated with outcome $Y$. However, I could not imagine the weights that are given by outcome $Y$. I believe that many common weighting schemes (including IPW) do not use $Y$, and I cannot imagine the weights based on $Y$. It would be clearer if the authors add the examples of outcome-based weights.

**Q5 Detailed Comments To The Authors:**

NA

**Q9 Complying With Reviewing Instructions:**

Yes

---

> ### Author Rebuttal · Authors · 2024-04-07
>
> We thank you for your time and valuable feedback!
>
> > Although the proposed weighting approach can deal with various treatment-effect statistics, the authors only evaluate the performance of average treatment effect (ATE) estimation in Section 5. I believe that the proposed method can address CATE estimation (according to Appendix ). If so, it would be better if the authors perform additional experiments.
>
> Let $P\_\text{obs}$ be the distribution of the observational data described in Section 2.1, such that any expectation $\mathbb{E}$ without indicating a distribution in subscript is wrt $P\_\text{obs}$, i.e. $\mathbb{E} := \mathbb{E}\_{P\_\text{obs}}$.
>
> Note that the framework of Problem 2.1 cannot directly address CATE estimation, as the target estimand $\mathbb{E}_Q[m(X)]$ is a scalar while the CATE is a function.
>
> Alternatively, one can perform multiple weightings as in Section 3.3, where for every problem we fix a covariate value $x_0$ and a treatment value $a$. Then we can take the pseudo-outcome to be $Y$, the source distribution $P$ to be $P\_{\text{obs}}(X,Y|A=a)$, the target distribution $Q$ to be $P\_{\text{obs}}(X|X=x\_0)$, thus the outcome model to be $m(x) = \mathbb{E}[Y | A=a, X=x]$. However, this choice of $Q$ would be a spike at $X=x_0$, potentially violating our assumption that $Q$ has a density in Section 2.2 depending on the base measure, e.g. if $X$ is continuous and the base measure is the Borel measure. As in [Ben-Michael et al. (2021)](https://arxiv.org/pdf/2110.14831.pdf), such a problem with spikes could be mitigated with smoothing, which is outside the scope of our submission. If $Q$ does have a density, e.g. when $X$ has categorical values and the base measure is the discrete uniform distribution, then it could actually be possible to perform weighting using our framework. However we are unaware of any other weighting paper in this setting.
>
> Thus, our framework does not naturally incorporate CATE estimation; we will revise the Appendix to make this clear. We will also revise the paragraph on generalizability and transportability in Appendix A, which might also lead to confusion. Here, while the CATE is the outcome model $m(X)$, the target estimand is not the CATE but the ATE on the whole population $\mathbb{E}[Y(1) - Y(0)]$ or on a separate population $\mathbb{E}[Y(1) - Y(0) | S=0]$.
>
> > In Section 1, the authors mention the importance of addressing « outcome-agnostic » weights, which are not formulated with outcome Y. However, I could not imagine the weights that are given by outcome Y. I believe that many common weighting schemes (including IPW) do not use Y, and I cannot imagine the weights based on Y. It would be clearer if the authors add the examples of outcome-based weights.
>
> This is a good point. Examples of using outcomes to derive weights include restricting the outcome function class, e.g. as a confidence interval around a regressed outcome model as in [Wainstein (2024)](https://arxiv.org/pdf/2203.12179.pdf), or deriving weights that are functions of the outcome, e.g. by estimating the density ratio between the source and target distributions of the outcome as in [Taufiq et al. (2023)](https://arxiv.org/pdf/2312.01457.pdf). Finally, many standard outcome modeling approaches, such as (kernel ridge) regression are implicitly weighting estimators so one could use such approaches to derive weights; see, for example, [Bruns-Smith et al. (2023)](https://arxiv.org/abs/2304.14545). We will further emphasize all of this in the manuscript.

---

### Official Review · Reviewer_rTjW · 2024-03-15

**Q2-1 Originality-Novelty:** 2
**Q2-2 Correctness-Technical Quality:** 3
**Q2-5 Clarity Of Writing:** 3

**Q1 Summary And Contributions:**

This paper explores theoretical properties and practical implementation of weighting estimators. There are two key contributions. Firstly, it analyzes the bias by decomposing it into three influential parts. Secondly, it proposes algorithms to find estimators that minimize each of these bias components.

**Q2-3 Extent To Which Claims Are Supported By Evidence:**

2: Fair: the main claims are somewhat supported by evidence (but the experimental evaluation may be weak, or does not match entirely with the claims, important baselines may be missing, proofs contain important ideas but lack rigor, algorithmic details are only discussed superficially, references are imprecise, assumptions are not sufficiently motivated or explicated, etc.).

**Q2-4 Reproducibility:**

2: Fair: key resources (e.g. proofs, code, data) are unavailable but key details (e.g. proof sketches, experimental setup) are sufficiently well-described for an expert to confidently reproduce the main results.

**Q3 Main Strengths:**

Section 3 offers insights into the mechanisms by which representation and weights influence bias. It also provides guidance to researchers on finding suitable representations.

**Q4 Main Weakness:**

1. This paper seems very similar to the paper “Towards representation learning for general weighting problems in causal inference” accepted to NeurIPS workshop 2023.
2. While this paper covers oracle properties and practical algorithms, it lacks a theoretical analysis of the proposed estimators. It might be better if the authors could give on discussions on whether the empirical estimator can achieve the oracle bound or not.
3. Additionally, there is typically a trade-off between bias and variance. Since the proposed method minimizes bias, is there a risk of it performing unsatisfactorily due to high variance?

**Q5 Detailed Comments To The Authors:**

1. The weight and its bias rely on the function class of the outcome model. In the proposed method, the function class is manually specified. Would there be a positive impact on bias bounds or estimation improvement if additional information about the function class were available?
2. If the function class M is mis-specified, would the proposed method perform better than traditional methods that use original X without imposing representations?
3. The authors compare their method with AutoDML. The DML approach strengthened on the inference of the parameter when state-of-art machine learning technique is applied in the first-stage estimation. Could there be some further discussion on inference of the proposed method?

**Q9 Complying With Reviewing Instructions:**

Yes

---

> ### Author Rebuttal · Authors · 2024-04-07
>
> We thank you for your time and valuable feedback!
>
> > Workshop paper
>
> We refer to the Program Chairs' comments.
>
> > Theoretical analysis, inference, AutoDML
>
> Assuming that $\text{Supp}(X) \subseteq \mathbb{R}^d$ is compact, $\text{Supp}(\tilde{Y}) \subseteq [0,1]$, $\tilde{w}\_{P,Q} \leq W\_{\text{max}}$ for some known $W\_{\text{max}}$, samples $\mathcal{P}$ and $\mathcal{Q}$ are disjoint, we outline a strategy to obtain confidence bounds for $\hat{\tau}\_{\hat{w}} - \tau$ where $\tau := \mathbb{E}\_Q[m(X)]$ and $\hat{\tau}\_{\hat{w}} = \frac{1}{\left|\mathcal{P}\right|} \sum\_{i \in \mathcal{P}} \hat{w}\_i \tilde{Y}\_i$ is the estimator where
> - $\hat{g}, \hat{\phi} \in \text{argmin}\_{\hat{g}, \hat{\phi} \ : \ \hat{g} \circ \hat{\phi} \in \mathcal{F}\_n} \hat{L}\_{\mathcal{P},\mathcal{Q}}(\hat{g} \circ \hat{\phi})$ where $\mathcal{F}\_n$ is a neural network architecture with $\text{Im}(\phi) \subseteq \mathbb{R}^{d'}$;
>
> - $\hat{w} = \text{argmin}\_{w \in [0, W\_{\text{max}}]^{\left|\mathcal{P}\right|}} \left|\left| \frac{1}{\left|\mathcal{P}\right|} \sum\_{i \in \mathcal{P}} w\_i k(., \hat{\phi}(x\_i)) - \frac{1}{\left|\mathcal{Q}\right|} \sum\_{i \in \mathcal{Q}} k(., \hat{\phi}(x\_i)) \right|\right|$ where $k$ is a bounded continuous kernel on $\mathbb{R}^{d'} \times \mathbb{R}^{d'}$.
>
> Then, $|\hat{\tau}\_{\hat{w}} - \tau| \leq |\hat{\tau}\_{\hat{w}} - \tau\_{\hat{\phi}}| + |\tau\_\hat{\phi} - \tau|$ where $\tau\_{\hat{\phi}} := \mathbb{E}\_Q[m^{P,\hat{\phi}}(\hat{\phi}(X))]$. We give a bound for these terms with probability $1 - \delta$:
> - For $|\hat{\tau}\_{\hat{w}} - \tau\_{\hat{\phi}}|$, we use [Yu and Szepesvari  (2012)](https://icml.cc/2012/papers/330.pdf) with $\hat{\phi}(X)$ instead of $X$ as the input space. Defining $\tilde{Q}\_{\hat{\phi}}(X,\tilde{Y}) := \frac{q(\hat{\phi}(X))}{p(\hat{\phi}(X))}P(X,\tilde{Y})$, we can take $P\_{\text{te}} = \tilde{Q}\_{\hat{\phi}}$ and $P\_{\text{tr}} = P$, then the setup and Assumptions 1-3 are verified. Thus we can use Theorems 2 and 3 if their $\mathcal{A}\_2$ or $\mathcal{A}\_\infty$ can be controlled using $\hat{\phi}$.
> - For $|\tau\_\hat{\phi} - \tau|$, our manuscript and $\tilde{Y} \in [0,1]$ lead to $|\tau\_\hat{\phi} - \tau| \leq ||\tilde{w}\_{P,Q} -  \hat{g} \circ \hat{\phi}||\_{L\_2(P)}$. We resort to [Chernozhukov et al. (2024)](https://arxiv.org/pdf/2104.14737.pdf). Let a binary $L$ and a distribution $R$ such that $R(L=1) = \frac{|\mathcal{Q}|}{|\mathcal{P}| + |\mathcal{Q}|}$, $R(X|L=1) = Q(X)$, $R(X|L=0) = P(X)$. Then, noting $W = (X,L,\tilde{Y})$ where we (artificially) extend $\tilde{Y}$ to $Q$, the functional can be taken as $m(W,\alpha) := \mathbb{E}\_R[\frac{L}{R(L=1)}\alpha(X,0)]$ as this retrieves $\tau = \theta(\gamma\_0) := \mathbb{E}\_R[m(W,\gamma\_0)]$ where $\gamma\_0(x,l) := \mathbb{E}[\tilde{Y} | X=x, L=l]$. One can show that $\theta$ has Riesz representer $\alpha\_0 = \bar{\alpha}(\tilde{w}\_{P,Q})$ where $\bar{\alpha}(v)(x,l) := \frac{1-l}{R(L=0)}v(x)$ and that $\bar{\alpha}(\hat{g} \circ \hat{\phi})$ is the solution of Equation (2.6) with $\mathcal{A}\_n = \{\bar{\alpha}(g \circ \phi), \ g \circ \phi \in \mathcal{F}\_n \}$. From $||\tilde{w}\_{P,Q} -  \hat{g} \circ \hat{\phi}||\_{L\_2(P)} = \sqrt{R(L=0)} ||\alpha\_0 - \bar{\alpha}(\hat{g} \circ \hat{\phi})||\_{L\_2(R)}$, as Assumption 1 is verified and Assumption 2 is probably verified too (Corollary 2.2 uses a similar $\mathcal{A}\_n$), Theorem 2.1 gives the confidence bound.
>
> Such results could be used to derive confidence intervals. However, unlike AutoDML our setup excludes outcome regression, so we cannot invoke doubly robustness.
>
> > trade-off between bias and variance.
>
> Our method incorporates a bias-variance tradeoff that can be decided through the $\sigma^2$ hyperparameter, as in e.g. [Ben-Michael et al. (2021)](https://arxiv.org/pdf/2110.14831.pdf).
>
> > if additional information about the function class
>
> Knowledge of the class $\mathcal{M}$ would ensure that bounds are valid for this specific $\mathcal{M}$. Knowledge about dependencies between covariates, treatment or outcome could also help choose a representation $\phi(X)$.
>
> > M mis-specified
>
> In our experiments, $m(x)$ is neither linear nor Sobolev of any order for $\mathbb{E}[Y(0)]$ in IHDP, for either $\mathbb{E}[Y(a)]$ in News, and for TBI. Thus, these $m(x)$ are misspecified for linear KMM (kernel mean matching) on all datasets and for energy balancing on IHDP and News (which have an odd input dimension, see page 12 of [Mak and Joseph (2018)](https://arxiv.org/pdf/1609.01811.pdf)). We conjecture that the $m(x)$ of TBI is misspecified for energy balancing too.
>
> For any given KMM method our representations $\hat{\phi}(X)$ outperforms covariates $X$ on News and TBI but are outperformed by them on IHDP. KMM with our $\hat{\phi}(X)$ also outperforms non-KMM baselines using $X$ directly (Entropy, IPW) more often than not (6 better performances vs. 3 ties and 3 worse performances, taking account of standard errors).

---

### Official Review · Reviewer_oimF · 2024-03-17

**Q2-1 Originality-Novelty:** 3
**Q2-2 Correctness-Technical Quality:** 3
**Q2-5 Clarity Of Writing:** 4

**Q10 Ethical Concerns:**

No.

**Q1 Summary And Contributions:**

The paper proposed a new characterization of bias in weighting problems based on representation learning. The upper bound on bias prompts a two-stage procedure for the learning of the representation and the weights. Numerical experiments show that representation learning achieves comparable bias to existing methods.

**Q2-3 Extent To Which Claims Are Supported By Evidence:**

4: Excellent: all claims are supported by very convincing evidence (in the form of comprehensive experimental evaluation, rigorous mathematical proofs, detailed (pseudo-)code, precise references, well-motivated and realistic assumptions) and the authors deliver what they promise.

**Q2-4 Reproducibility:**

3: Good: key resources (e.g. proofs, code, data) are available and key details (e.g. proofs, experimental setup) are sufficiently well-described for competent researchers to confidently reproduce the main results.

**Q3 Main Strengths:**

The work does a good job tying together many weight-based causal inference problems. The bias in such problems can be exposed using any representation of the target function to be reweighted. Representation learning can be combined with AutoDML to produce estimators with low biases.

**Q4 Main Weakness:**

The learning of representations appears to be merely an intermediate step to obtain the final estimate. But in the second stage, one still has to assume a function class for the target function, this time wrt to the representation learnt. The need for representation learning is not well motivated.

**Q5 Detailed Comments To The Authors:**

1. In the second column on page 2, the definition of the source and target distributions do not include the (pseudo-)outcome. This may be somewhat confusing, since the quantity $m(X)$ itself is defined using the joint distribution $P(\tilde{Y},X)$.
2. In the second to last paragraph on page 4, Supplement 2.4 appears to refer to Appendix B.
3. If one learn the function $v$ as a composition of two functions $g\circ \phi$, the functions $g$ and $\phi$ will not be uniquely identifiable in general. How does this affect the estimation of weights in the subsequent stage?
4. In the section for numerical results, the acronym "PCA" probably refers to principle component analysis, but it is not defined.
5. An important application of the method imho is in the handling of multimodal data through the use of neural network, such as medical scans and texts.
6. Can the learnt representations be used for other purposes? For example, if they can be seen as deconfounding scores, can one also "interpret" them as confounding strength and use these in sensitivity analysis for unmeasured confounding?
---
7. In the paragraph below Proposition 3.3, the word order is wrong in the sentence "the second term of the RHS, find plug [...] and weights minimizing it".
8. There are two right parentheses in $||\tilde{w}_{P,Q}(X)-v(X){\color{red})}||$.
9. In the second column on page 8, the formula for the joint bias is in the wrong place with a trailing comma.
10. The use of -ize and -ise endings is inconsistent.

**Q9 Complying With Reviewing Instructions:**

Yes

---

> ### Author Rebuttal · Authors · 2024-04-07
>
> We thank you for your time and valuable feedback!
>
> > The need for representation learning is not well motivated.
>
> We agree that the motivation was insufficient in the initial submission and have tried to strengthen it in the revision. Some motivations for using a representation even as an input to an IPM whose function class is assumed are :
> - The use of common IPMs such as the Wasserstein distance or the MMD for any kernel might not be appropriate in high dimensions, as illustrated by Dudley (1969) and Ramdas et al. (2015) referenced in our paper, which justifies using a low-dimensional representation to feed to these distances. The better performance of low-dimensional representations compared to the original covariate space for both kernel mean matching on the linear kernel and the energy distance kernel in our experiments further illustrates this motivation.
> - Learning such representations might help select the most relevant information about the DGP, e.g. pertaining to confounding, which then might improve the efficiency of the balancing weights method used in the second step, as suggested by work on variable selection for treatment effect estimation, e.g. in Brookhart et al. (2006) which we reference as well.
>
> > definition of the source and target distributions
>
> > Supplement 2.4 / Appendix B.
>
> > PCA not defined.
>
> > An important application of the method imho is in the handling of multimodal data through the use of neural network, such as medical scans and texts.
>
> > "the second term of the RHS, find plug [...] and weights minimizing it".
>
> > There are two right parentheses in $||\tilde{w}_{P,Q}(X) - v(X))||$
>
> > the formula for the joint bias is in the wrong place with a trailing comma.
>
> > The use of -ize and -ise endings is inconsistent.
>
> We thank you for pointing out these confusions, typos and remarks which we will address or incorporate in our manuscript.
>
>
> > If one learn the function $v$ as a composition of two functions $g \circ \phi$, the functions $g$ and $\phi$ will not be uniquely identifiable in general. How does this affect the estimation of weights in the subsequent stage?
>
> It is true that without restrictions many different $(g, \phi)$ tuples will share the same value of the AutoDML loss, e.g. any $(g_h, \phi_h) = (g \circ h, h^{-1}(\phi))$ where $h$ is invertible. However, restricting $g$ and $\phi$ to be components of a neural network with a given architecture will exclude many possible invertible $h$'s. Some $h$'s will remain though, such as $h(z) = \lambda \odot z$ where $\odot$ is the Hadamard product and $\lambda_i \neq 0 \ \forall i$, which means that the returned $\phi$ might have arbitrary amplitude or smoothness, which would influence the weighting stage if e.g. the IPM is made considerably smaller or larger wrt the $\sigma^2 \cdot ||w||_{L^2(P)}$ term in the equation of Proposition 3.1. A workaround could be in adding some regularization of $\phi$ in the AutoDML loss, eg through weight decay. We do not perform weight decay and still obtain competitive performance, which suggests that Adam optimization might choose an appropriate $\phi$ in practice.
>
>
>
> > Can the learnt representations be used for other purposes?
>
> Our method could be of use in [Johansson et al. (2019)](https://arxiv.org/pdf/1903.03448.pdf). Indeed, the "excess target information loss" $\eta_{\phi}^\ell(f,Y)$ in this paper is equal to the confounding bias defined in Equation (2) of our manuscript, where $P := P_s$, $Q := P_t$, $m(x) := m(x;\phi)$ where $m(x;z) := \mathbb{E}[\ell(f(z(X), Y)) | X=x]$. The assumptions of Theorem 3 in Johansson et al. (2019) then ensure that $\|m(.,z)\|\_{L_2(P_s)} \leq M$ for some $M > 0$. Thus, using our Proposition 3.2 and the Cauchy-Schwarz inequality, we have that $\eta_{\phi}^\ell(f,Y) \leq M \cdot \text{BSE}\_{P_s,P_t}(\phi)$ and $M \cdot \text{BSE}\_{P_s,P_t}(\phi)$ can replace $\eta_{\phi}^\ell(f,Y)$ in the formulae of Theorem 3 in Johansson et al. (2019).
>
> However, representations learnt using our method are generally not deconfounding scores, unless they make the BSE zero or make the confounding bias zero more generally, the latter of which is not verifiable in the absence of outcome information.
>
> Sensitivity analysis might be orthogonal to our setting, as our manuscript's confounding bias pertains to the relationship between $X$ and $\phi(X)$, while the confounding strength in sensitivity analysis pertains to the relationship between $X$ and unobserved confounders $U$. Using a representation $\phi(X)$ instead of original covariates $X$ could potentially facilitate some calculations, however.
>
>
> > However, the numerical experiments were not designed to show the strengths of the two-stage method.
>
> We do think that the numerical experiments show competitive performance of our method : the best method using our representation is only outperformed by methods using NSM on News, by IPW on TBI and is within one standard error of the best performing methods on IHDP.

---

### Official Review · Reviewer_aySZ · 2024-03-19

**Q2-1 Originality-Novelty:** 3
**Q2-2 Correctness-Technical Quality:** 3
**Q2-5 Clarity Of Writing:** 3

**Q1 Summary And Contributions:**

The paper aims to find a weight function for reweighting the original distribution to a target distribution for generalizing causal effects, e.g., average treatment effect. The approach involves learning a representation of the covariate space. The main contribution is quantifying the information loss due to using an estimator based on representation (instead of the original covariates) and analiyzing the bias. An algorithm, inspirred by the prioer work, is also proposed to learn such representations.

**Q2-3 Extent To Which Claims Are Supported By Evidence:**

3: Good: the main claims are supported by convincing evidence (in the form of adequate experimental evaluation, proofs, (pseudo-)code, references, assumptions).

**Q2-4 Reproducibility:**

3: Good: key resources (e.g. proofs, code, data) are available and key details (e.g. proofs, experimental setup) are sufficiently well-described for competent researchers to confidently reproduce the main results.

**Q3 Main Strengths:**

- The paper is generally well-written with a clear problem statement, assumptions, intuitions, and connection to related work.
- The bias is carefully analyzed, which sheds light on the role of the learned representation.
- The algorithm is fairly clear and demonstrates stronger performance than the baseline methods.

**Q4 Main Weakness:**

- The jump from confounding bias to balancing score error can be too coarse. Surely, it addresses many of the issues regarding not knowing the ground-truth models. That being said, some analysis of specific cases or intuition for the gap between them would be useful.
- As discussed by the authors in conclusion, assuming a canonical RKHS for the function class $\cal M$ can be limiting.

**Q5 Detailed Comments To The Authors:**

Providing more intuition/discussion regarding the points in weaknesses (especially the first one) would be helpful.

**Q9 Complying With Reviewing Instructions:**

Yes

---

> ### Author Rebuttal · Authors · 2024-04-07
>
> We thank you for your time and valuable feedback!
>
> > The jump from confounding bias to balancing score error can be too coarse. Surely, it addresses many of the issues regarding not knowing the ground-truth models. That being said, some analysis of specific cases or intuition for the gap between them would be useful.
>
> An important class of examples is that the confounding bias will be zero and the BSE will be non-negative, even potentially arbitrary, for many representations $\phi(x)$ that incorporate information on the outcome model $m(x)$. Concrete examples include prognostic scores from [Hansen (2008)](https://www.jstor.org/stable/20441477), or the deconfounding scores in the example of Section 5 in [D'Amour and Franks (2021)](https://arxiv.org/pdf/2104.05762.pdf). This is the price of discarding outcome information - either for computational convenience or by assumption as in our setting.
>
> However, note that the confounding bias is non-trivial to evaluate for an arbitrary representation $\phi$, even for known outcome model $m(x)$ and true weights $\tilde{w}_\text{P,Q}(x)$, as it requires the computation of the exact conditional expectation of both functions wrt the representation variable $\phi(X)$. Thus, the gap will generally be difficult to assess outside strong assumptions on the data generating process and the form of $\phi$, e.g. as in Section 5 of D'Amour and Franks (2021) who assume Gaussian covariates, linear representations, and generalized linear models for the outcome model and propensity score (thus true weights). We expect such an assessment to be significant further work.
>
> > As discussed by the authors in conclusion, assuming a canonical RKHS for the function class $\mathcal{M}$ can be limiting.
>
> In the conclusion, we refer to the challenge of charactersing the class $\phi(\mathcal{M}, P)$ (defined in Equation (3)) of the projected outcome model $m^{P, \phi}(\varphi) = \mathbb{E}_P[m(X) | \phi(X) = \varphi]$ depending on the class $\mathcal{M}$ of the original outcome model $m(x)$.
>
> At the time of submitting, we had no insight on this and were only considering assuming canonical RKHSs for $\phi(\mathcal{M}, P)$.  However, since submission, we have found results in [Clivio et al. (2022)](https://proceedings.mlr.press/v151/clivio22a/clivio22a.pdf) that yield a first characterisation. For example, from their Proposition 9, if $\mathcal{M}$ is a class of $K$-Lipschitz functions and $\phi(x)$ is a neural network with invertible and $K'$-bi-Lipschitz activation functions, then $\phi(\mathcal{M}, P)$ will be included in the class of Lipschitz functions with a Lipschitz constant $K''$ that depends on $K, K'$ and the weights of $\phi$. Although this result does not help yet when $\mathcal{M}$ is a kernel as we would like for the convenient kernel mean matching method, we will include it in our manuscript.

---

### Official Review · Reviewer_3JFe · 2024-03-22

**Q2-1 Originality-Novelty:** 2
**Q2-2 Correctness-Technical Quality:** 3
**Q2-5 Clarity Of Writing:** 3

**Q1 Summary And Contributions:**

The paper proposes an end-to-end estimation procedure that learns a flexible representation, while retaining promising theoretical properties. The authors show that this approach is competitive in a range of common causal inference tasks.

**Q2-3 Extent To Which Claims Are Supported By Evidence:**

3: Good: the main claims are supported by convincing evidence (in the form of adequate experimental evaluation, proofs, (pseudo-)code, references, assumptions).

**Q2-4 Reproducibility:**

3: Good: key resources (e.g. proofs, code, data) are available and key details (e.g. proofs, experimental setup) are sufficiently well-described for competent researchers to confidently reproduce the main results.

**Q3 Main Strengths:**

1. The paper is technically sound.
2. The theory looks strong.

**Q4 Main Weakness:**

The paper could be more clear in notation.

**Q5 Detailed Comments To The Authors:**

1. In Eq(2), I don't see why $LHS = E_{P_w}[m^{P_w,\phi}(\phi(X))] - E_Q[m^{Q,\phi}(\phi(X))]$, could you provide a derivation? And it would be helpful to give a definition for $m^{R,\phi}(X)$ where R is a distribution.
2. After Proposition 3.1, the author uses the example of $E[Y(a)]$ to explain the confounding bias. Again, I don't see why $m^{Q,\phi}(\phi(X))$ is $E[E[Y|A=a,X]]$ in this case.

**Q9 Complying With Reviewing Instructions:**

Yes

---

> ### Author Rebuttal · Authors · 2024-04-07
>
> We thank you for your time and valuable feedback!
>
> > The paper could be more clear in notation.
>
> We thank you for pointing this out. We hope that our answers to your questions make all of this clearer - we will incorporate these into the manuscript and see how we can improve the overall clarity of notations.
>
> > In Eq(2), I don't see why $\text{LHS} = E\_{P\_w}[m^{P\_w,\phi}(\phi(X))] - E\_{Q}[m^{Q,\phi}(\phi(X))]$, could you provide a derivation? And it would be helpful to give a definition for $m^{R,\phi}(X)$ where R is a distribution.
>
> Taking the notation from the RHS, and expanding expectations, we have
>
> $\text{Bias wrt the representation} = \underbrace{E\_{P\_w}[m^{P,\phi}(\phi(X))]}\_{ (A)} - \underbrace{E\_{Q}[m^{P,\phi}(\phi(X))]}\_{(B)}$
>
> $\text{Chosen weights bias} = \underbrace{E\_{P\_w}[m^{P\_w,\phi}(\phi(X))]}\_{(C)} - \underbrace{E\_{P\_w}[m^{P,\phi}(\phi(X))]}\_{(D)}$
>
> $\text{Confounding bias} = \underbrace{E\_{Q}[m^{P,\phi}(\phi(X))]}\_{(E)} - \underbrace{E\_{Q}[m^{Q,\phi}(\phi(X))]}\_{(F)}$
>
> Summing these three biases, $(A)$ and $(D)$ cancel out, and so do $(B)$ and $(E)$. Thus, we obtain $\text{LHS} = (C) - (F)$, that is exactly $\text{LHS} = E\_{P\_w}[m^{P\_w,\phi}(\phi(X))] - E\_{Q}[m^{Q,\phi}(\phi(X))]$.
>
> As for $m^{R,\phi}(X)$, we assume that you mean $m^{R,\phi}(\phi(X))$ as you do in the next question ($m^{R,\phi}$ is not defined on the space of $X$ but that of $\phi(X)$). From the first equation in Section 3.1, $m^{R,\phi}(\phi(X)) = E\_R[m(X) \mid \phi(X)]$.
>
> For completeness, note that from the tower property, $E\_R[m^{R,\phi}(\phi(X))] = E\_R[E\_R[m(X) \mid \phi(X)]] = E\_R[m(X)]$. Applied to $R = P\_w$ and $R = Q$, this yields $\text{LHS} = \mathbb{E}\_{P\_w}[m(X)] - \mathbb{E}\_Q[m(X)] = \text{Bias}\_{P,Q}(w;m)$, as in the paper.
>
> > After Proposition 3.1, the author uses the example of $E[Y(a)]$ to explain the confounding bias. Again, I don't see why $m^{Q,\phi}(\phi(X))$ is $E[E[Y|A=a,X]]$ in this case.
>
> Let $P\_\text{obs}$ be the distribution of the observational data described in Section 2.1, such that all expectations in $E[E[Y | A=a, X]]$ and generally any $E$ without indicating a distribution in subscript are wrt $P\_\text{obs}$, i.e. $E := E\_{P\_\text{obs}}$. Then weighting to obtain $E[Y(a)]$ corresponds to Problem 2.1 with $\tilde{Y} := Y$, $P := P\_\text{obs}(X, Y \mid A = a)$, $Q := P\_\text{obs}(X)$, thus $m(x) := E[Y | X=x, A=a]$. Note that here $E\_Q[m(X)] = E[E[Y | X, A=a]] = E[Y(a)]$ where the last equality comes from canonical identification assumptions in the potential outcomes framework such as unconfoundedness, overlap and consistency.
>
> Then, we do recover the fact that the target estimand $E\_Q[m(X)]$ is precisely both $E[Y(a)]$ (from identification assumptions) and the $E[E[Y | X, A=a]]$ term which you raise (by definition). It is not equal to $m^{Q,\phi}(\phi(X))$ however, but to $\mathbb{E}\_Q[m^{Q,\phi}(\phi(X))]$ due to the tower property as explained in the former answer.
>
> Thus, the $(F)$ term of the confounding bias as expanded above is equal to $E[E[Y \mid X, A=a]]$. For completeness, we now show that the $(E)$ term is equal to $E[E[Y \mid \phi(X), A=a]]$, which would justify our description of the confounding bias given for $\mathbb{E}[Y(a)]$ after Proposition 3.1, which you refer to. By definition,
>
> $m^{P,\phi}(\phi(X)) = E_P[m(X) | \phi(X)] =  E[E[Y \mid X, A=a] \mid \phi(X), A=a]$.
>
> From the fact that knowing $X$ implies knowing $\phi(X)$,
>
> $m^{P,\phi}(\phi(X)) =  E[E[Y \mid X, \phi(X), A=a] \mid \phi(X), A=a] $
>
> thus, from the tower property, we have
>
> $m^{P,\phi}(\phi(X)) = E[ Y \mid \phi(X), A=a]$.
>
> Taking the expectation in $E\_Q$ and knowing that $E\_Q = E\_{P\_\text{obs}} = E$ gives $(E) = E\_{Q}[m^{P,\phi}(\phi(X))] = E\_{Q}[E[Y | \phi(X), A=a]] = E[E[Y | \phi(X), A=a]]$, which completes the proof. We hope that all of this has made the part about the confounding bias for $E[Y(a)]$ clearer.

---

### Meta-Review · Area_Chair_uWkM · 2024-04-22

This paper proposes a weighting approach for treatment effect estimation. It theoretically analyzes the bias term and explicitly formulates the confounding bias term, which is used for estimating a better representation for treatment effect estimation. The proposed method is theoretically sound.

Overall, reviewers appreciate the technical solidity and well-crafted writing. Please also note that the paper would be stronger if there were richer experimental results to further support the theoretical claims.